# Regulatory DNA sequence Design with Reinforcement Learning

**Zhao Yang[1]*  Bing Su[1]†  Chuan Cao[2]†  Ji-Rong Wen[1]**

[1]Gaoling School of Artificial Intelligence, Renmin University of China

[2]Microsoft Research AI4Science

## Abstract

*Cis*-regulatory elements (CREs), such as promoters and enhancers, are relatively short DNA sequences that directly regulate gene expression. The fitness of CREs, measured by their ability to modulate gene expression, highly depends on the nucleotide sequences, especially specific motifs known as transcription factor binding sites (TFBSs). Designing high-fitness CREs is crucial for therapeutic and bioengineering applications. Current CRE design methods are limited by two major drawbacks: (1) they typically rely on iterative optimization strategies that modify existing sequences and are prone to local optima, and (2) they lack the guidance of biological prior knowledge in sequence optimization. In this paper, we address these limitations by proposing a generative approach that leverages reinforcement learning (RL) to fine-tune a pre-trained autoregressive (AR) model. Our method incorporates data-driven biological priors by deriving computational inference-based rewards that simulate the addition of activator TFBSs and removal of repressor TFBSs, which are then integrated into the RL process. We evaluate our method on promoter design tasks in two yeast media conditions and enhancer design tasks for three human cell types, demonstrating its ability to generate high-fitness CREs while maintaining sequence diversity. The code is available at https://github.com/yangzhao1230/TACO.

## 1 Introduction

*Cis*-regulatory elements (CREs), such as promoters and enhancers, are short functional DNA sequences that regulate gene expression in a cell-type-specific manner (Fu et al., 2025). Promoters determine when and where a gene is activated, while enhancers boost gene expression levels. Over the past decade, millions of putative CREs (Gao & Qian, 2020) have been identified, but these naturally evolved sequences only represent a small fraction of the possible genetic landscape and are not necessarily optimal for specific expression outcomes. It is crucial to design synthetic CREs with desired fitness (measured by their ability to enhance gene expression) as they have broad applications in areas such as gene therapy (Boye et al., 2013), synthetic biology (Shao et al., 2024), precision medicine (Collins & Varmus, 2015), and agricultural biotechnology (Gao, 2018).

Previous attempts to explore alternative CREs have relied heavily on directed evolution, which involves iterative cycles of mutation and selection in wet-lab settings (Wittkopp & Kalay, 2012; Heinz et al., 2015). This approach is sub-optimal due to the vastness of the DNA sequence space and the significant time and cost required for experimental validation. For example, a 200 base pair (bp) DNA sequence can have up to $2.58 \times 10^{120}$ possible combinations (Gosai et al., 2024), far exceeding the number of atoms in the observable universe. Thus, efficient computational algorithms are needed to narrow down the design space and prioritize candidates for wet-lab testing.

Massively parallel reporter assays (MPRAs) (de Boer et al., 2020) have enabled the screening of large libraries of DNA sequences and their fitness in specific cell types. Recent studies have begun using fitness prediction models as reward models to guide CRE optimization, allowing the explo-

---

*Work was done during internship at Microsoft Research AI4Science: yangyz1230@gmail.com

†Correspondence to: Bing Su (subingats@gmail.com), Chuan Cao (chuancao.926@gmail.com)

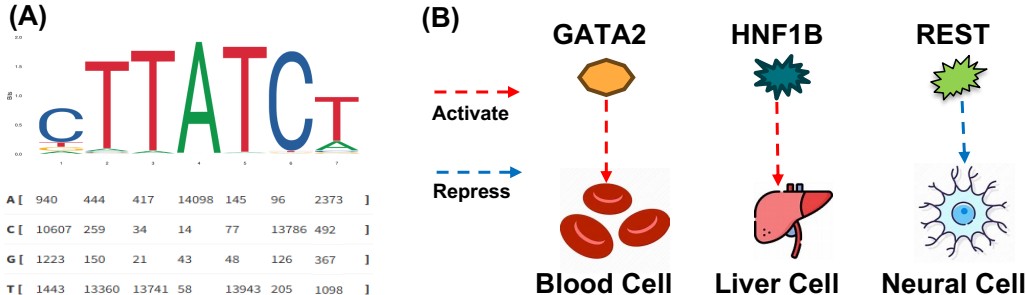

Figure 1: (A) TFBSs are commonly represented as frequency matrices, indicating the frequency of each nucleotide appearing at specific positions within the binding site. (B) GATA2 and HNF1B specifically activate gene expression in blood cells and liver cells, respectively, while REST specifically represses gene expression in neural cells.

ration of sequences that outperform natural ones (Vaishnav et al., 2022; de Almeida et al., 2024). These methods typically rely on iterative optimization strategies that modify existing or random sequences. In each iteration, they modify previously selected sequences to generate new candidates. Although the search space is vast, these methods can only explore a limited local neighborhood through simple modifications, making it difficult for them to escape local optima. As a result, these approaches often produce CREs with limited diversity. Moreover, these methods generally use generic sequence optimization techniques without incorporating biological prior knowledge.

Inspired by the success of using Reinforcement Learning (RL) to fine-tune autoregressive (AR) generative language models (Ouyang et al., 2022; Liu et al., 2024; Mo et al., 2024), we propose a generative approach that leverages RL to fine-tune AR models for designing cell-type-specific CREs. Unlike previous methods that modify existing sequences, our approach enables the generation of novel sequences from scratch by capturing the underlying distribution of CREs. We employ HyenaDNA (Nguyen et al., 2024b; Lal et al., 2024), a state-of-the-art (SOTA) AR DNA generative model, and fine-tune it on CREs to learn their natural sequence patterns, ensuring the generation of realistic and diverse sequences. During RL fine-tuning, we treat the current AR model as the policy network and use the fitness predicted by a reward model as the reward signal to update our policy. This allows us to adjust the model parameters to generate CRE sequences that achieve high fitness while maintaining sequence diversity.

Additionally, we seamlessly incorporate domain knowledge of CREs into our RL process. The regulatory syntax of CREs is largely dictated by the transcription factors (TFs) that bind to them (Gosai et al., 2024; de Almeida et al., 2024; Lal et al., 2024; Zhang et al., 2023). TFs are proteins that directly influence gene expression by binding to specific sequence motifs within CREs, known as TF binding sites (TFBSs), and modulating transcriptional activity. For instance, Figure 1 (A) shows the motif pattern recognized by the *GATA2* TF. Furthermore, the effects of TFs can vary widely depending on the cell type. As shown in Figure 1 (B), *GATA2* and *HNF1B* are TFs that specifically activate gene expression in blood cells and liver cells (Lal et al., 2024), respectively, while *REST* acts as a repressor of gene expression in neural cells (Zullo et al., 2019), illustrating the cell-type-specific nature of TF activity.

The effect of a TF can be broken down into its intrinsic role as an activator or repressor (de Almeida et al., 2024) (referred to as its *vocabulary*) and its interactions with other TFs (Georgakopoulos-Soares et al., 2023). We find that using the frequency of TFBS occurrences within a sequence as features can yield reasonably good fitness prediction performance when trained with a decision tree model, LightGBM (Ke et al., 2017). As shown in Table 1, the SOTA DNA model, Enformer, achieves a Pearson correlation of 0.83 on the test set for predicting fitness in the HepG2 cell line using se-

| Model | yeast | | human | | |
|---|---|---|---|---|---|
| | complex | defined | hepg2 | k562 | sknsh |
| Enformer (Sequence Feature) | 0.87 | 0.91 | 0.83 | 0.85 | 0.85 |
| LightGBM (TFBS Frequency Feature) | 0.63 | 0.65 | 0.65 | 0.65 | 0.66 |

Table 1: Pearson correlation coefficient of different fitness prediction models on the test set. Details can be found in Section 3.3.

quence data as input. In contrast, using only TFBS frequency features (without sequence information) results in a Pearson correlation of 0.65. This shows that TF frequency alone can still capture much of the predictive power, even without detailed sequence data. Notably, we believe that these frequency features implicitly reflect both the TF vocabulary and their interactions. Furthermore, we use the trained LightGBM (Ke et al., 2017) model to infer whether each TFBS feature promotes or represses fitness using SHAP values (Lundberg, 2017), allowing us to explicitly integrate TFBS domain knowledge into our RL process. Hence, we name our proposed method **TACO**: **T**FBS-**A**ware *Cis*-Regulatory Element **O**ptimization, which combines RL fine-tuning of autoregressive models with domain knowledge of TFBSs to enhance CRE optimization.

Our main contributions are as follows:

- We introduce the RL fine-tuning paradigm for pre-trained AR DNA models in CRE design, enabling the generated sequences to maintain high diversity while also exploring those with superior functional performance.
- We incorporate TFBS information by inferring their regulatory roles and integrating their impact directly into the generation process, allowing for a synergistic exploration guided by both data-driven and knowledge-driven approaches.
- We evaluate our approach under different optimization settings on real-world datasets, including yeast promoter designs from two media types and human enhancer designs from three cell lines, demonstrating the effectiveness of TACO.

## 2 RELATED WORK

**Conditional DNA Generative Models**. DDSM (Avdeyev et al., 2023) pioneered diffusion models as a generative approach for DNA design, using classifier-free guidance (Ho & Salimans, 2022) to generate DNA sequences based on desired promoter expression levels. Building on this work, several studies have since employed diffusion models for CRE design (Li et al., 2024b; DaSilva et al., 2024; Sarkar et al., 2024). In addition to diffusion models, regLM (Lal et al., 2024) utilized prefix-tuning on the AR DNA language model HyenaDNA (Nguyen et al., 2024b), incorporating tokens that encode expression strength to fine-tune the model specifically for CRE design. However, these generative methods are primarily designed to fit existing data distributions, which limits their ability to design novel sequences with potentially higher fitness levels that have yet to be explored by humans.

**DNA Sequence Optimization**. DyNA PPO (Angermueller et al., 2019) was an early exploration of applying modern RL to biological sequence design. By improving the sampling efficiency of PPO (Schulman et al., 2017) and leveraging an AR policy, it provided a general framework for biological sequence design. DyNA PPO and its subsequent works (Jain et al., 2022; Zeng et al., 2024) primarily focused on advancing general-purpose sequence design algorithms, with an emphasis on optimizing short TFBS motifs (6-8 bp) in the context of DNA sequence design. With the availability of larger CRE fitness datasets, Vaishnav et al. (2022) applied genetic algorithms to design CREs. Recent works, such as Gosai et al. (2024), explored greedy approaches like AdaLead (Sinai et al., 2020), simulated annealing (Van Laarhoven et al., 1987), and gradient-based SeqProp (Linder & Seelig, 2021). Similarly, Taskiran et al. (2024) combined greedy strategies with directed evolution. However, these methods often start from random sequences or observed high-fitness sequences, leading to local optima and limited diversity. Recently, Reddy et al. (2024) proposed directly optimizing CREs using gradient ascent (GAs) on a differentiable reward model trained on offline CRE datasets.

**Motif-based Machine Learning in Scientific Data**. Motifs are small but critical elements in scientific data, such as functional groups in molecules or TFBS in DNA sequences. Explicitly modeling these motifs can provide significant benefits, as demonstrated in molecular optimization (Jin et al., 2020; Chen et al., 2021), generation (Geng et al., 2023), property prediction (Zhang et al., 2021), and DNA language models (An et al., 2022). In the context of DNA CREs, TFBSs are widely considered the most important motifs. Our approach is heavily inspired by de Almeida et al. (2024), who observed that during direct evolution guided by a reward model, there is a tendency to first remove repressor TFBSs and then add enhancer TFBSs to optimize the sequences. We have incorporated this observed process into our RL framework.

## 3 METHOD

### 3.1 PROBLEM FORMULATION

We define a DNA sequence $X = (x_1, \cdots, x_L)$ as a sequence of nucleotides with length $L$, where $x_i \in \{A, C, G, T\}$ is the nucleotide at the $i$-th position. We assume the availability of a large-scale dataset of CRE sequences with fitness measurements $\mathcal{D} = \{(X_1, f(X_1)), \cdots, (X_N, f(X_N))\}$, where $N$ is the number of sequences in the dataset and $f(X)$ represents the fitness for $X$. On the complete dataset $\mathcal{D}$, we train a reward model *oracle* to predict a CRE's fitness, which is used for the final evaluation. Additionally, we partition a subset of low-fitness sequences $\mathcal{D}_{\text{low}} \subset \mathcal{D}$ to train both an AR model and a reward model *surrogate* that reflects the distribution learned from the offline dataset. Different experimental settings may utilize different types of reward models to guide the RL optimization process if applicable. Our ultimate goal is to generate sequences predicted to have high fitness by the oracle.

To align with RL terminology, we formulate sequence generation as a Markov Decision Process (MDP). The state $s_i$ corresponds to the partial sequence generated up to time step $i$, while the action $a_i \in \{A, C, G, T\}$ represents the nucleotide selected at position $i$. Our generative model serves as the policy $\pi_\theta$, which learns to output a probability distribution over possible actions (nucleotides) given the current state (partial sequence). The generation process terminates when the sequence reaches length $L$.

### 3.2 RL-BASED FINE-TUNING FOR AUTOREGRESSIVE DNA MODELS

**Pre-training CRE-specific AR Model.** We adapt HyenaDNA (Nguyen et al., 2024b) for our AR model by continual training on $\mathcal{D}_{\text{low}}$ following Lal et al. (2024). While HyenaDNA achieves strong performance on DNA tasks with linear complexity (Appendix C), it was originally trained on the whole human genome rather than short regulatory CRE sequences. To bridge this length and functional gap, we perform continual pre-training to better capture CRE-specific regulatory patterns (Experimental evidence in Appendix C.1). We refer to this process as pre-training since it employs unsupervised learning on the sequence data.

The pre-trained AR model serves as our initial policy $\pi_\theta$ in the RL framework, predicting the probability of each nucleotide given previous ones. We formulate this as a sequential process where each action $a_i \in \{A, C, G, T\}$ selects a nucleotide at position $i$. The policy is trained by minimizing:

$$\min_\theta \mathbb{E}_{x \sim \mathcal{D}_{\text{low}}} \left[ \sum_{i=1}^{L} -\log \pi_\theta(a_i \mid a_1, \cdots, a_{i-1}) \right], \tag{1}$$

where $\pi_\theta(a_i \mid a_1, \cdots, a_{i-1})$ represents the probability assigned by the policy to selecting nucleotide $a_i$ at position $i$ given the sequence of previous actions (nucleotides). Pre-training on $\mathcal{D}_{\text{low}}$ helps the policy learn to generate sequences that already resemble the true CRE distribution (Jin et al., 2020; Chen et al., 2021), providing a good initialization for subsequent RL fine-tuning.

**RL-Based Fine-tuning for AR DNA Models**. Next, we formulate the RL fine-tuning process as a MDP, as illustrated in Figure 2. In this formulation, the state $s_i$ corresponds to the partial sequence generated up to time step $i$, while the action $a_i$ represents the nucleotide selected by the policy $\pi_\theta$. The reward $r(s_{i-1}, a_i)$ is defined as a combination of two types of rewards: TFBS reward $r_{\text{TFBS}}$ and fitness reward $r_{\text{fitness}}$, as shown in equation 2:

$$r(s_{i-1}, a_i) = \begin{cases} r_{\text{fitness}}, & \text{if } i = L, \\ r_{\text{TFBS}}(t), & \text{if } a_i \text{ results in a TFBS } t \in \mathcal{T}, \\ 0, & \text{otherwise.} \end{cases} \tag{2}$$

Here, $r_{\text{fitness}}$ is applied when $i$ is the final time step of the episode ($i = L$), and represents the fitness value of the generated sequence as evaluated by the reward model. On the other hand, $r_{\text{TFBS}}$ is a reward applied whenever a TFBS $t \in \mathcal{T} = \{t_1, t_2, t_3, \ldots, t_n\}$ is identified in the sequence after selecting $a_i$. Details on how TFBSs are identified can be found in Appendix D. The specific values of $r_{\text{TFBS}}(t)$ are introduced in Section 3.3. Negative rewards are assigned for generating repressive TFBSs, while positive rewards are given for generating activating TFBSs, as shown in Figure 2. The overall objective is to maximize the expected cumulative reward:

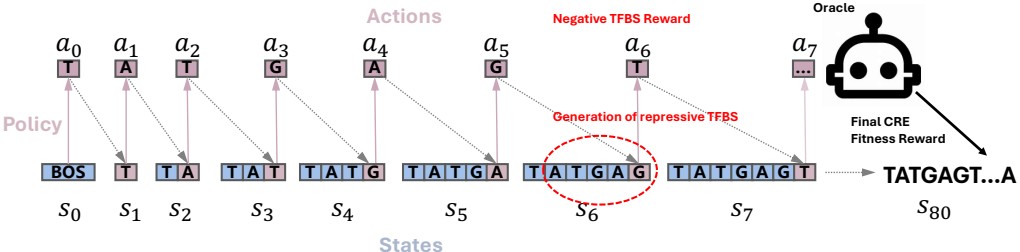

Figure 2: **AR generation of a DNA sequence.** The action $a_i$ represents the nucleotide to be appended to the sequence, and the state $s_{i-1}$ is the concatenation of all actions taken up to time $i-1$. A negative reward is given if an action generates a repressive TFBS, while a positive reward is given for an activating TFBS. The final sequence is evaluated using a reward model (oracle) to obtain its fitness. BOS represents the beginning of the sequence, and *ATCG* denotes the nucleotide bases.

$$\max_\theta J(\theta) = \mathbb{E}_{\pi_\theta} \left[ \sum_{i=1}^{L} r(s_{i-1}, a_i) \right], \tag{3}$$

where $J(\theta)$ represents the expected cumulative reward, $L$ is the length of the sequence, and $r(s_{i-1}, a_i)$ is the reward at each step. This objective enables the policy to generate DNA sequences with desired regulatory properties by leveraging both reward model guidance and domain-specific knowledge of TFBS vocabulary.

**Auxiliary RL Techniques**. To optimize the policy $\pi_\theta$, we employ the REINFORCE algorithm (Williams, 1992) following previous studies in molecule optimization (Ghugare et al., 2024). Additionally, we leverage a hill climbing replay buffer (Blaschke et al., 2020), which stores and samples high-fitness sequences during training to further guide exploration. We also apply entropy regularization (Ghugare et al., 2024), which inversely weights the log probabilities of selected actions. This approach effectively penalizes actions with excessively high probabilities, thereby discouraging overconfident actions and promoting exploration of less likely ones. This combination of techniques allows the model to effectively balance exploration and exploitation, resulting in improved performance on complex DNA optimization tasks. Ablation experiments supporting these designs can be found in Appendix G.2.

### 3.3 INFERENCE OF TFBS REGULATORY ROLES

As illustrated in Figure 3, our approach to inferring TFBS regulatory roles consists of two steps. First, we train a decision tree-based fitness prediction model using TFBS frequency features as input. Second, we leverage the model interpretability technique SHAP values (Lundberg, 2017) to determine the regulatory impact of each TFBS feature.

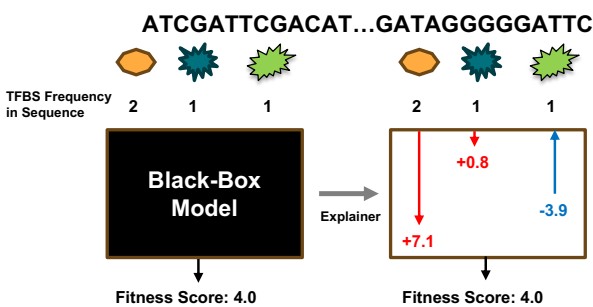

Figure 3: A black-box LightGBM model takes TFBS occurrences as input, and SHAP values infer their contributions to CRE fitness prediction.

To infer the regulatory impact of each TFBS, we first define the TFBS frequency feature of a sequence $x$ as a vector $\mathbf{h}(X) = [h_1(X), h_2(X), \ldots, h_n(X)]$, where $h_i(X)$ denotes the frequency of the $i$-th TFBS in sequence $x$. This feature vector represents the occurrence pattern of TFBSs within the sequence, making it suitable for tabular data modeling. Details on extracting TFBS features by scanning the sequence can be found in Appendix D. Given the tabular nature of this data, we employ LightGBM (Ke et al., 2017), a tree-based

model known for its interpretability and performance on tabular datasets, to fit the fitness values of sequences. LightGBM is chosen because decision tree models, in general, offer better interpretability by breaking down the contribution of each feature in a clear, hierarchical manner. Details of the LightGBM model can be found in Appendix E. After training, we evaluate the model's performance using the Pearson correlation coefficient between the true and predicted fitness values, as shown in Table 1. This evaluation metric helps us quantify how well the LightGBM model captures the relationship between TFBS frequencies and fitness values.

Based on the trained LightGBM model, we use SHAP values (Lundberg, 2017) to interpret the impact of each TFBS on the predicted fitness. SHAP values provide a theoretically grounded approach to attribute the prediction of a model to its input features by calculating the contribution of each feature (in our case, each TFBS) to the prediction. The SHAP value for the $i$-th TFBS in sequence $X$, denoted as $\phi_i(X)$, is computed as:

$$\phi_i(X) = \sum_{S \subseteq \{1,\dots,n\} \setminus \{i\}} \frac{|S|!(n - |S| - 1)!}{n!} \left( \hat{f}(S \cup \{i\}) - \hat{f}(S) \right), \tag{4}$$

where $S$ is a subset of features not containing $i$, $\hat{f}(S \cup \{i\})$ is the model prediction when feature $i$ is included, and $\hat{f}(S)$ is the prediction when feature $i$ is excluded. This equation ensures that SHAP values fairly distribute the impact of each feature according to its contribution. To infer the reward $r_{\text{TFBS}}(t)$ for each TFBS $t \in \mathcal{T} = \{t_1, t_2, t_3, \dots, t_n\}$, we compute the mean SHAP value of $t$ over the entire dataset. If the mean SHAP value does not significantly differ from zero (p-value $> 0.05$, determined by hypothesis testing), we set the reward of $t$ to zero:

$$r_{\text{TFBS}}(t) = \begin{cases} \alpha \cdot \mu_\phi(t), & \text{if } p\text{-value} < 0.05, \\ 0, & \text{otherwise}, \end{cases} \tag{5}$$

where $\alpha$ is a tunable hyperparameter, and $\mu_\phi(t)$ is the mean SHAP value of TFBS $t$ across the dataset. This approach ensures that only statistically significant TFBSs contribute to the reward, and $\alpha$ controls the magnitude of the reward. Analysis of cell-type-specific TFBSs based on SHAP values is provided in Appendix D.1.

### 3.4 Summary of Our Method

In summary, our method integrates two key components. First, we fine-tune an AR generative model, pre-trained on CRE sequences, using RL to optimize sequence generation (Figure 2). Second, we use a data-driven approach to infer the role of TFBSs in a cell-type-specific context and incorporate these insights into the RL process (Figure 3). The complete workflow is detailed in Appendix G Algorithm 1.

## 4 Experiment

### 4.1 Experiment Setup

**Datasets**. We conducted experiments on both yeast promoter and human enhancer datasets. The yeast promoter dataset includes two types of growth media: *complex* (de Boer et al., 2020) and *defined* (Vaishnav et al., 2022). The human enhancer dataset consists of three cell lines: HepG2, K562, and SK-N-SH (Gosai et al., 2024). All paired CRE sequences and their corresponding fitness measurements were obtained from MPRAs (Sharon et al., 2012). Our data preprocessing was based on Lal et al. (2024) (Appendix A). The DNA sequence length in the yeast promoter dataset is 80, while it is 200 for the human enhancer dataset. Each dataset represents a cell-type-specific scenario due to distinct TF effect vocabularies and regulatory landscapes.

**Reward Model and Pre-training Dataset.** In Section 4.2 and Section 4.3, we show experiments under different settings. In Section 4.2, we assume access to an ideal oracle that guides the RL process, where both guidance and evaluation utilize oracle scoring. The policy $\pi_\theta$, however, is pre-trained exclusively on $\mathcal{D}_{\text{low}}$. In contrast, Section 4.3 presents an offline Model-Based Optimization

(MBO) setting, which assumes no access to the ideal oracle during the optimization process. Instead, we rely solely on a surrogate model trained on $\mathcal{D}_{\text{low}}$ for scoring, while the policy is pre-trained on the complete dataset $\mathcal{D}$. Only after generating the final batch of candidate sequences do we submit them to the oracle for evaluation. Architecture details of the reward models are provided in Appendix B.

**Baselines**. We compare TACO against several established optimization approaches, including Bayesian optimization as implemented in the FLEXS benchmark (Sinai et al., 2020), and evolutionary algorithms such as AdaLead (Sinai et al., 2020) and PEX (Anand & Achim, 2022), as well as covariance matrix adaptation evolution strategy (CMAES) (Auger & Hansen, 2012) using one-hot encoding. Additionally, we adapt the SOTA protein optimization method LatProtRL (Lee et al., 2024) for CRE optimization. Given the lack of a powerful backbone model like ESM (Jain et al., 2022) in the DNA domain (Evidence is provided in Appendix C.2), we removed the ESM-based latent vector encoding from LatProtRL and refer to the resulting model as DNARL. DNARL can be viewed as a sequence mutation-based PPO algorithm (Schulman et al., 2017) enhanced with a replay buffer mechanism. We do not compare with Gradient Ascent (GAs)-based approaches (Reddy et al., 2024), as our method and other baselines do not rely on differentiable surrogates. For a detailed discussion on GAs, please refer to Appendix J.

**Evaluation Metrics.** We use three evaluation metrics: *Top*, *Medium*, and *Diversity*. *Top* is defined as the mean fitness value of the highest-performing 16 sequences from the optimized set $\mathcal{X}^* = \{X_1, \dots, X_K\}$, where $K = 256$ represents the total number of sequences proposed in each round. Both *Medium* and *Diversity* metrics are calculated using the highest-performing 128 sequences selected from the full set of 256 sequences. *Medium* refers to the median fitness value among these 128 sequences, while *Diversity* quantifies the median pairwise distance between all pairs within these 128 sequences. These metrics align with those used in Lee et al. (2024), except for *Novelty*, which we omit because DNA sequences lack well-defined validity criteria, resulting in disproportionately high novelty scores with limited informative value. For details, see Appendix F. We report the mean and standard deviation of all evaluation metrics across five runs with different random seeds. Standard deviations appear in parentheses in these tables.

**Implementation Details**. The architecture of our AR model is based on HyenaDNA-1M[1], following the approach outlined in Lal et al. (2024). In Section 4.2, we pre-trained all policies on the subset $D_{\text{low}}$. In Section 4.3, we directly used regLM as the initial policy, but without adding prefix tokens to simulate unsupervised training (Lal et al., 2024). All experiments were conducted on a single NVIDIA A100 GPU. During optimization, we set the learning rate to 5e-4 for the yeast task and 1e-4 for the human task. The hyperparameter $\alpha$, which controls the strength of the TFBS reward in equation 5, was set to 0.01. We min-max normalized all reported fitness values and the rewards used for updating the policy, while the reward models were trained on the original fitness values. For each round, we generated $K = 256$ sequences, each with a fixed length, consistent with the datasets. A total of $E = 100$ optimization rounds were conducted, and the metrics were computed using the final round's $\mathcal{X}^*$.

## 4.2 FITNESS OPTIMIZATION WITH AN ACTIVE LEARNING SETTING

In this setting, we assume that the oracle trained on the complete dataset $D$ is accessible during the RL process. Following Lee et al. (2024); Ghugare et al. (2024), we partitioned each dataset into three subsets—*easy*, *middle*, and *hard*—based on fitness values, denoted as $D_{\text{low}}$. Detailed partitioning strategies are provided in Appendix A. For each difficulty level, we pre-trained the AR model on $D_{\text{low}}$ to simulate optimization starting from low-fitness sequences.

**Yeast Promoters**. As shown in Table 2, optimizing yeast promoters is relatively easy, with most methods generating sequences that significantly exceed the dataset's maximum observed fitness values. For such sequences, the results are reported as **1**. Therefore, we only present the results for the hard subset, while the complete results are available in Appendix

| Method | Yeast Promoter (Complex) | | | Yeast Promoter (Defined) | | |
|---|---|---|---|---|---|---|
| | Top | Medium | Diversity | Top | Medium | Diversity |
| PEX | 1 | 1 | 9.8 (1.48) | 1 | 1 | 9.8 (2.59) |
| AdaLead | 1 | 1 | 7.6 (0.89) | 1 | 1 | 6.4 (0.55) |
| BO | 1 | 1 | 5.6 (5.57) | 1 | 1 | 5.6 (1.04) |
| CMAES | 0.79 (0.02) | 1 | 30.0 (2.5) | 0.44 (0.03) | 1 | 30.4 (2.3) |
| DNARL | 1 | 1 | 7.7 (0.48) | 1 | 1 | 10.2 (1.4) |
| TACO | 1 | 1 | **52.8 (2.77)** | 1 | 1 | **49.6 (3.65)** |

Table 2: Results on yeast promoter datasets (hard).

---

[1]https://huggingface.co/LongSafari/hyenadna-large-1m-seqlen-hf

| Method | HepG2-easy | | | HepG2-middle | | | HepG2-hard | | |
|---|---|---|---|---|---|---|---|---|---|
| | Top | Medium | Diversity | Top | Medium | Diversity | Top | Medium | Diversity |
| PEX | **0.93** (0.02) | **0.89** (0.01) | 20.2 (6.57) | **0.89** (0.04) | **0.86** (0.04) | 19.2 (7.12) | **0.85** (0.04) | **0.82** (0.02) | 16.0 (2.65) |
| AdaLead | 0.76 (0.00) | 0.75 (0.00) | 5.2 (0.45) | 0.75 (0.03) | 0.74 (0.03) | 12.4 (4.04) | 0.74 (0.02) | 0.73 (0.02) | 8.0 (1.87) |
| BO | 0.66 (0.06) | 0.60 (0.09) | 41.6 (8.91) | 0.63 (0.05) | 0.58 (0.05) | 42.0 (7.81) | 0.68 (0.04) | 0.63 (0.08) | 39.8 (5.07) |
| CMAES | 0.61 (0.06) | 0.42 (0.04) | 77.4 (4.04) | 0.67 (0.02) | 0.43 (0.03) | 75.0 (3.24) | 0.69 (0.03) | 0.43 (0.02) | 77.2 (5.17) |
| DNARL | 0.79 (0.07) | 0.71 (0.02) | 12.2 (0.08) | 0.63 (0.14) | 0.84 (0.09) | 7.32 (0.01) | 0.76 (0.04) | 0.72 (0.01) | 20.0 (3.42) |
| TACO | 0.78 (0.01) | 0.75 (0.01) | **131.8** (2.39) | 0.76 (0.01) | 0.73 (0.01) | **139.4** (7.13) | 0.76 (0.01) | 0.74 (0.01) | **131.8** (4.27) |

| Method | K562-easy | | | K562-middle | | | K562-hard | | |
|---|---|---|---|---|---|---|---|---|---|
| | Top | Medium | Diversity | Top | Medium | Diversity | Top | Medium | Diversity |
| PEX | **0.95** (0.01) | **0.93** (0.01) | 21.8 (9.68) | 0.94 (0.01) | **0.92** (0.01) | 14.6 (1.82) | **0.95** (0.01) | **0.92** (0.02) | 15.9 (1.34) |
| AdaLead | 0.85 (0.01) | 0.84 (0.01) | 7.0 (1.00) | 0.85 (0.01) | 0.84 (0.01) | 9.0 (1.87) | 0.85 (0.01) | 0.84 (0.01) | 8.8 (1.64) |
| BO | 0.70 (0.13) | 0.65 (0.12) | 41.6 (5.32) | 0.76 (0.05) | 0.70 (0.05) | 39.6 (5.55) | 0.74 (0.03) | 0.70 (0.04) | 37.0 (6.52) |
| CMAES | 0.70 (0.05) | 0.42 (0.02) | 78.8 (4.09) | 0.79 (0.03) | 0.50 (0.03) | 76.0 (3.24) | 0.73 (0.05) | 0.47 (0.05) | 76.8 (4.55) |
| DNARL | 0.89 (0.04) | 0.87 (0.01) | 23.3 (3.72) | 0.90 (0.02) | 0.86 (0.01) | 26.3 (1.88) | 0.89 (0.01) | 0.87 (0.02) | 17.5 (3.33) |
| TACO | 0.93 (0.00) | 0.91 (0.01) | **124.6** (3.51) | 0.92 (0.01) | 0.90 (0.02) | **126.0** (1.58) | 0.93 (0.01) | 0.91 (0.01) | **125.6** (2.88) |

| Method | SK-N-SH-easy | | | SK-N-SH-middle | | | SK-N-SH-hard | | |
|---|---|---|---|---|---|---|---|---|---|
| | Top | Medium | Diversity | Top | Medium | Diversity | Top | Medium | Diversity |
| PEX | 0.90 (0.01) | 0.86 (0.03) | 22.2 (5.93) | **0.92** (0.02) | **0.88** (0.01) | 23.8 (7.85) | 0.90 (0.02) | 0.86 (0.03) | 23.0 (2.74) |
| AdaLead | 0.84 (0.08) | 0.82 (0.08) | 7.4 (1.52) | 0.81 (0.06) | 0.80 (0.06) | 9.4 (3.05) | 0.79 (0.05) | 0.78 (0.05) | 14.4 (4.45) |
| BO | 0.68 (0.07) | 0.62 (0.07) | 39.8 (7.89) | 0.71 (0.08) | 0.64 (0.10) | 40.4 (4.83) | 0.71 (0.06) | 0.63 (0.04) | 39.9 (6.60) |
| CMAES | 0.73 (0.04) | 0.45 (0.02) | 77.0 (3.39) | 0.74 (0.01) | 0.45 (0.03) | 76.0 (3.81) | 0.74 (0.02) | 0.44 (0.03) | 76.0 (3.54) |
| DNARL | 0.83 (0.21) | 0.80 (0.06) | 35.42 (2.99) | 0.83 (0.01) | 0.81 (0.01) | 28.8 (1.93) | 0.82 (0.01) | 0.81 (0.01) | 18.7 (3.21) |
| TACO | **0.91** (0.01) | **0.87** (0.02) | **133.8** (4.27) | 0.90 (0.01) | 0.86 (0.01) | **135.0** (2.12) | **0.92** (0.00) | **0.88** (0.01) | **137.4** (1.14) |

Table 3: Performance comparison on human enhancer datasets.

Table 14. Among the baselines, only CMAES fails to fully optimize sequences to the maximum fitness value, although it demonstrates strong performance in terms of diversity. Our method not only achieves the maximum fitness but also exhibits the highest diversity compared to all other approaches.

**Human Enhancers**. Optimizing human enhancers presents a more challenging task. As shown in Appendix A Table 6, the 90th percentile min-max normalized fitness values for HepG2, K562, and SK-N-SH in the dataset $D$ are 0.4547, 0.4541, and 0.4453, respectively—less than half of the maximum observed. In Table 3, TACO demonstrates superior performance compared to the baselines. For the HepG2, PEX achieves the highest fitness score, but its diversity is typically below 20. In contrast, TACO attains SOTA fitness for K562 and SK-N-SH cell lines while maintaining significantly higher diversity across all datasets (over 1/3 higher than CMAES, which has the highest diversity among baselines).

**Evaluation by Optimization Round**. As shown in Figure 4, we present the evaluation results after each round of optimization. We observe that AdaLead, a greedy-based algorithm, quickly finds relatively high-fitness sequences at the initial stages. However, its diversity drops rapidly, causing the fitness to plateau and get stuck in local optima. In contrast, PEX demonstrates a steady increase in fitness, but it consistently maintains a low diversity throughout. Only TACO not only achieves a stable increase in fitness but also maintains high diversity due to its generative model paradigm.

## 4.3 Offline Model-Based Optimization

We present results of offline MBO (Reddy et al., 2024; Uehara et al., 2024), where the dataset $\mathcal{D}$ is partitioned into a subset $D_{\text{offline}}$. The AR model was pre-trained on $\mathcal{D}$ to simulate real-world scenarios where sequences are abundant but their fitness labels are unknown (Uehara et al., 2024). Meanwhile, we trained a fitness prediction model (surrogate) on the smaller $D_{\text{offline}}$ to guide the RL process. The maximum fitness threshold in $D_{\text{offline}}$ is defined as the 95th percentile of $\mathcal{D}$. Unlike Section 4.2, where the true oracle is known, here we lack such knowledge. Details of why we need offline MBO and data partitioning strategies are provided in Appendix H. Since the oracle is not visible during the optimization process, we introduce an additional evaluation metric: the average pairwise cosine similarity of the embeddings of the proposed sequences as generated by the oracle

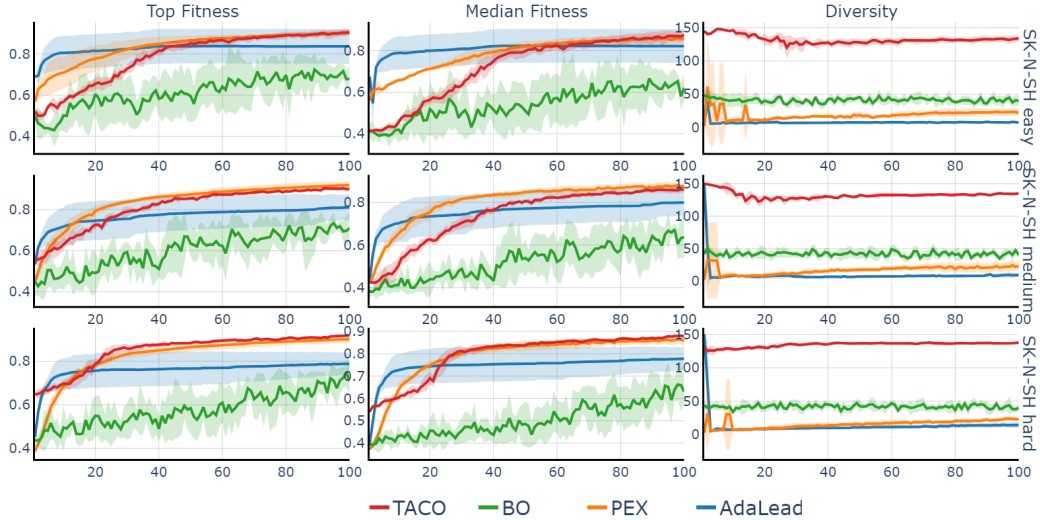

Figure 4: **Evaluation metrics by optimization round** for TACO, BO, PEX and Adalead. Shaded regions indicate the standard deviation of 5 runs. The x-axis indicates the number of rounds.

model. This metric, referred to as *Emb Similarity*, quantifies the diversity of the final proposed sequences in the latent featre space.

Table 4 presents the results on the K562 dataset. Under the offline MBO, the performance of all methods degrades compared to the oracle-guided setting. The overall trends across methods are consistent with those observed in Section 4.2. TACO achieves results in Top and Median that are comparable to PEX while significantly outperforming other optimization methods in terms of Diversity. The complete offline MBO results for all datasets are presented in Appendix K. The results of lowering the fitness threshold for the offline dataset are presented in Appendix J. We also include two conditional generative models, regLM (Lal et al., 2024) and DDSM (Avdeyev et al., 2023) These methods maintain high diversity in the generated sequences; however, the fitness of the generated sequences is generally inferior to that achieved by most optimization methods. See detailed discussion and implementation details of generative models in Appendix I.

| Model | Top ↑ | Medium ↑ | Diversity ↑ | Emb Similarity ↓ |
|---|---|---|---|---|
| PEX | **0.76** (0.02) | **0.73** (0.02) | 15.8 (4.97) | 0.97 (0.01) |
| AdaLead | 0.66 (0.08) | 0.58 (0.06) | 63.2 (70.01) | 0.88 (0.12) |
| BO | 0.71 (0.07) | 0.64 (0.08) | 43.6 (6.91) | 0.87 (0.04) |
| CMAES | 0.66 (0.02) | 0.44 (0.03) | 79.2 (3.83) | **0.35** (0.03) |
| reglm | 0.69 (0.02) | 0.47 (0.01) | **149.60** (0.49) | 0.38 (0.02) |
| DDSM | 0.43 (0.00) | 0.40 (0.00) | 93.40 (0.49) | 0.80 (0.00) |
| TACO | 0.75 (0.09) | 0.72 (0.10) | 102.6 (20.14) | 0.97 (0.04) |

Table 4: Offline MBO results for human enhancers (K562).

## 4.4 ABLATION STUDY

Fine-tuning a pre-trained AR model with RL and incorporating the TFBS reward are our key contributions. We conducted ablation experiments in the setting described in Section 4.3. Results are shown in Table 5.

**The effect of pre-training.** Pre-training on CRE sequences proves to be highly beneficial. While the "w/o pretraining" setup (which uses a randomly initialized policy) occasionally discovers sequences with high fitness, it underperforms on the Medium metric by 0.03, 0.12, and 0.03 compared to the

second-best result across datasets. This demonstrates that pre-training allows the policy to begin in a relatively reasonable exploration space, enabling it to identify a large number of suitable sequences more efficiently.

**The effect of TFBS reward.** Incorporating the TFBS reward enhances the Medium metric of TACO. The method outperforms the "w/o TFBS reward" baseline by margins of 0.02, 0.01, and 0.02, respectively. These prior-informed rewards guide the policy to explore a more rational sequence space efficiently. Moreover, the biologically guided TFBS reward is surrogate-agnostic, with the potential to achieve a similar effect to the regularization applied to surrogates in Reddy et al. (2024), by avoiding excessive optimization towards regions where the surrogate model gives unusually high predictions. The differences in the Top and Diversity achieved by various models are relatively minor, with no consistent conclusion. As the $\alpha$ increases from the default value of 0.01 to 0.1, our method shows improved performance in both Top and Medium metrics for K562 and SK-N-SH datasets. However, this improvement comes at the cost of a rapid drop in diversity. Interestingly, all metrics for the HepG2 dataset worsen as $\alpha$ grows. We hypothesize that this discrepancy arises from the quality of TFBS reward, precomputed using the LightGBM model, varying across datasets. We recommend carefully tuning $\alpha$ in real-world scenarios.

| Dataset | Setting | Top ↑ | Medium ↑ | Diversity ↑ | Emb Similarity ↓ |
|---|---|---|---|---|---|
| HepG2 | TACO ($\alpha = 0.01$) | **0.69** (0.03) | **0.60** (0.05) | **141.2** (1.92) | 0.82 (0.05) |
| | w/o rre-training | 0.68 (0.00) | 0.55 (0.02) | 139.4 (2.30) | 0.69 (0.02) |
| | w/o TFBS reward | 0.66 (0.05) | 0.58 (0.07) | 140.8 (1.64) | **0.81** (0.05) |
| | $\alpha = 0.1$ | 0.65 (0.06) | 0.58 (0.06) | 138.6 (3.21) | 0.86 (0.04) |
| K562 | TACO ($\alpha = 0.01$) | 0.75 (0.09) | 0.72 (0.10) | 102.6 (20.14) | 0.97 (0.04) |
| | w/o pre-training | 0.66 (0.15) | 0.59 (0.16) | **103.6** (25.77) | **0.83** (0.14) |
| | w/o TFBS reward | 0.76 (0.07) | 0.71 (0.08) | 106.2 (20.90) | 0.94 (0.05) |
| | $\alpha = 0.1$ | **0.78** (0.01) | **0.77** (0.01) | 82.8 (4.02) | 0.99 (0.00) |
| SK-N-SH | TACO ($\alpha = 0.01$) | 0.68 (0.08) | 0.62 (0.08) | 121.4 (7.86) | 0.90 (0.03) |
| | w/o pre-training | 0.69 (0.02) | 0.57 (0.06) | **131.8** (11.17) | **0.74** (0.11) |
| | w/o TFBS reward | 0.67 (0.06) | 0.60 (0.06) | 111.6 (12.86) | 0.89 (0.04) |
| | $\alpha = 0.1$ | **0.71** (0.01) | **0.65** (0.02) | 121.2 (5.45) | 0.90 (0.05) |

Table 5: Ablation study on the effect of pre-training and TFBS reward.

## 5 LIMITATIONS

There are still several areas for improvement in our approach: (1) The TFBS candidates we use are derived from a fixed database, which bounds the upper limit of the TFBS reward. Exploring data-driven motif mining (Dudnyk et al., 2024) methods may help. (2) Currently, we infer the role of TFs based solely on TFBS frequency. In reality, interactions between TFs and their orientation can significantly impact their regulatory roles (Georgakopoulos-Soares et al., 2023). Explicitly incorporating these factors to model more complex TF activities could lead to further improvements. (3) Evaluating DNA sequence validity requires further attention. Unlike for molecules or proteins, defining valid DNA sequences is more challenging, necessitating additional metrics beyond trained reward models alone (Appendix F).

## 6 CONCLUSION

Designing CREs is a highly impactful task with increasing data availability. We propose TACO, a generative approach that leverages RL to fine-tune a pre-trained autoregressive model. TACO incorporates biological priors through additional rewards for activator TFBS addition and repressor TFBS removal during the RL process. Our method generates high-fitness CREs while maintaining sequence diversity. To our knowledge, we are among the first to explicitly integrate TFBS motif information into machine learning-based CRE design. We believe this approach of incorporating such fundamental biological knowledge into generative models represents a promising direction for future research in the field.

## ACKNOWLEDGEMENTS

This work was supported in part by the National Natural Science Foundation of China No. 62376277 and No. 61976206, and Doubao large model fund.

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

# A    DETAILS OF DATASETS

Existing CRE fitness datasets were generated using Massively Parallel Reporter Assays (MPRAs), which enable high-throughput measurements of regulatory sequences. The yeast promoter dataset includes results from two distinct media conditions: *complex* (de Boer et al., 2020) and *defined* (Vaishnav et al., 2022). In contrast, the human enhancer dataset consists of data from three different human cell lines (Gosai et al., 2024): HepG2 (a liver cell line), K562 (an erythrocyte cell line), and SK-N-SH (a neuroblastoma cell line). The fitness differences in human enhancers are substantial, which makes the optimization task more challenging. As shown in Table 6, the 90th percentile min-max normalized fitness values for HepG2, K562, and SK-N-SH in dataset $D$ are 0.4547, 0.4541, and 0.4453, respectively.

| Cell Line | 75th Percentile | 90th Percentile |
|---|---|---|
| HepG2 | 0.3994 | 0.4547 |
| K562 | 0.3975 | 0.4541 |
| SK-N-SH | 0.3986 | 0.4453 |

Table 6: Enhancer fitness.

In Section 4.2, we adopted the dataset splits proposed by regLM (Lal et al., 2024), using their defined training set as our full dataset, denoted as $\mathcal{D}$. To simulate a progression from low-fitness to high-fitness sequences, we further partitioned $\mathcal{D}$ into a subset $\mathcal{D}_{\text{low}}$ for pre-training the policy. Specifically, we define three difficulty levels—*hard*, *medium*, and *easy*—based on fitness percentiles of 20–40, 40–60, and 60–80, respectively, for both media conditions in the yeast dataset. Since yeast is a single-cell organism, we ensured that fitness levels remained consistent across both media. For the human enhancer datasets, we define the *hard* fitness range as values below 0.2, the *medium* range as values between 0.2 and 0.75, and the *easy* range as values between 0.75 and 2.5. These thresholds were chosen to maintain fitness values below 0.2 in other cell lines, thereby simulating a cell-type-specific regulatory scenario. The selected sequences within these fitness ranges form $\mathcal{D}_{\text{low}}$, which is used for pre-training the policy.

# B    ENFORMER SERVES AS REWARD MODELS (ORACLE AND SURROGATE)

Enformer (Avsec et al., 2021) is a hybrid architecture that integrates CNNs and Transformers, achieving SOTA performance across a range of DNA regulatory prediction tasks. In our study, all CRE fitness prediction models are based on the Enformer architecture (Lal et al., 2024; Uehara et al., 2024). The primary distinction lies in the output: while the original Enformer model predicts 5,313 human chromatin profiles, we adapted it to predict a single scalar value (Lal et al., 2024) representing CRE fitness.

The reward models for the human enhancer datasets retain the same number of parameters as the original Enformer. In contrast, for the yeast promoter datasets, we reduced the model size due to the simpler nature of yeast promoter sequences, following Lal et al. (2024). Specific architectural configurations are detailed in Table 7. For consistency, we directly utilized the pre-trained weights provided by regLM (Lal et al., 2024) as our oracle. However, we independently trained our own surrogate model on $D_{\text{offline}}$.

| Model | Dimension | Depth | Number of Downsamples |
|---|---|---|---|
| Human Enhancer | 1536 | 11 | 7 |
| Yeast Promoter | 384 | 1 | 3 |

Table 7: Reward model hyperparameters.

## C    DISCUSSION ON DNA FOUNDATION MODELS

Over the past year, there has been significant growth in the development of DNA foundation models, with many new models emerging. However, most of these models, such as Caduceus (Schiff et al.), DNABert2 (Zhou et al., 2024), and VQDNA (Li et al., 2024a), are based on BERT-style pretraining and lack the capability to generate DNA sequences. Among them, HyenaDNA (Nguyen et al., 2024b) is the only GPT-style DNA language model. Unlike traditional Transformer-based architectures, HyenaDNA leverages a state space model (SSM), which provides linear computational complexity, making it suitable for handling long DNA sequences with complex dependencies. Subsequent work based on HyenaDNA, such as Evo (Nguyen et al., 2024a), has demonstrated the powerful DNA sequence generation capabilities of this architecture. Additionally, regLM (Lal et al., 2024) has explored conditional DNA generation by employing a prefix-tuning strategy, where a customized token is used as the prefix of the DNA sequence to guide the subsequent generation process. This approach has enabled reglm to effectively model context-dependent DNA sequence generation.

### C.1    EFFECT OF PRE-TRAINING ON SHORT CREs

Although HyenaDNA can be directly employed as an initial policy, its pre-training was conducted on 1M-length sequences across the entire human genome. Therefore, as described in Section 3.2, we further fine-tuned the initial oracle on CRE datasets. As shown in Table 8, fine-tuning HyenaDNA on short functional CRE sequences leads to modest performance improvements. We attribute these enhancements to the fine-tuning process, which exposes the model to a greater number of short sequences, thereby better aligning it with the sequence lengths required for subsequent CRE design tasks. Furthermore, this adaptation enhances the model's ability to capture functional regions specific to CREs.

| Model | Top ↑ | Medium ↑ |
|---|---|---|
| Pre-trained HyenaDNA | 0.749 | 0.723 |
| Fine-tuned HyenaDNA | **0.751** | **0.729** |

Table 8: Performance (hepg2 hard) comparison of pre-trained and fine-tuned HyenaDNA on short CRE sequences.

### C.2    LIMITATIONS OF CURRENT DNA FOUNDATION MDOELS

While there have been advancements in DNA foundation models, evidence suggests that they do not yet match the capabilities of models like ESM (Vaishnav et al., 2022). Specifically: (1) ESM embeddings are known for their high versatility and are widely utilized in various downstream tasks, e.g., enzyme function prediction (Yu et al., 2023). In contrast, as noted in Tang & Koo (2024), DNA foundation model embeddings often **perform no better than one-hot encodings**. (2) ESM's language model head can achieve AUROC scores above 0.9 in pathogenic mutation prediction by directly calculating the log-likelihood ratio of reference and alternative alleles (Meier et al., 2021). However, DNA foundation models currently perform significantly worse, with AUROC scores below 0.6 as reported in Benegas et al. (2023). (3) In addition to sequence-based DNA foundation models, some supervised DNA models have also been shown to exhibit limitations in distinguishing mutations across individuals Huang et al. (2023) and recognizing long-range DNA interactions Karollus et al. (2023).

## D    TFBS SCAN AND FREQUENCY FEATURE PREPROCESSING

The Jaspar database (Fornes et al., 2020) provides detailed annotations of TFBSs. Each TFBS $t_i$ corresponds to a transcription factor that binds to it, regulating gene expression. Instead of representing $t_i$ as a fixed sequence, it is described by a position frequency matrix $\mathbf{M}_i \in \mathbb{R}^{L_i \times 4}$, where $L_i$ is the length of the TFBS, and the four columns correspond to the nucleotides $\{A, C, G, T\}$. The matrix encodes the likelihood of each nucleotide appearing at each position in the TFBS, making it possible to capture variations in TF binding.

We utilize FIMO (Find Individual Motif Occurrences) (Bailey et al., 2015) to scan each sequence for potential TFBSs. Given a sequence $x$ and a matrix $\mathbf{M}_i$, FIMO evaluates each subsequence $x_j$ in $x$ by calculating a probabilistic score:

$$\text{score}(x_j, \mathbf{M}_i) = \prod_{k=1}^{L_i} P(n_k \mid \mathbf{M}_i[k]), \tag{6}$$

where $P(n_k \mid \mathbf{M}_i[k])$ represents the probability of nucleotide $n_k$ occurring at position $k$ in the matrix $\mathbf{M}_i$. FIMO identifies the subsequences with the highest scores as potential occurrences of the TFBS.

For each sequence $X$, FIMO outputs a frequency feature vector $\mathbf{h}(x) = [h_1(X), h_2(X), \ldots, h_n(X)]$ , where $\mathbf{h}_i(x)$ denotes the frequency of the $i$-th TFBS in sequence $x$. This frequency feature vector is then used as input for the downstream prediction model. The use of frequency-based features, as opposed to binary indicators, captures the varying levels of TFBS occurrences in the sequence, allowing for a more nuanced understanding of the regulatory role of each TFBS. Given this tabular representation, we employ LightGBM (Ke et al., 2017), a tree-based model known for its interpretability and effectiveness on tabular datasets, to predict the fitness values of sequences.

### D.1 Cell-type Specificity of TFBS Functional Roles

After analyzing the contribution of each TFBS using SHAP values (Section 3.3), we further investigated whether TFBSs exhibit cell-type specific regulatory functions. Figure 5 and Figure 6 present Venn diagrams illustrating the functional classification of TFBSs across different conditions in yeast and human datasets, respectively. For yeast promoters (Figure 5), we observed that TFBSs maintain

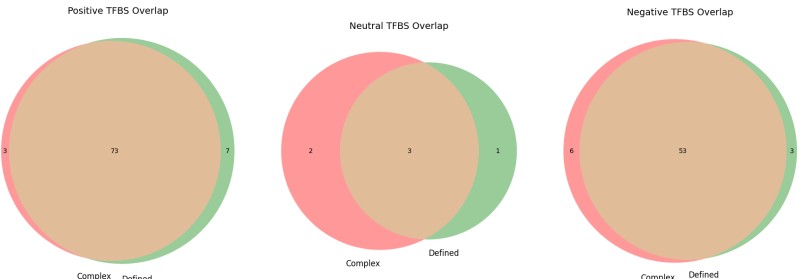

Figure 5: Venn diagram categorizing TFBSs by their functional roles (positive, neutral, or negative) in yeast promoters across two media conditions (Complex and Defined). The substantial overlap across all categories indicates that TFBSs maintain consistent regulatory functions regardless of media composition, confirming the absence of condition-specific TFBS activity within the same cell type.

consistent functional roles between Complex and Defined media conditions. This is evidenced by the substantial overlap in the Venn diagrams for positive, neutral, and negative regulatory categories. Since both conditions represent the same cell type with only differing media composition, this result confirms that TFBS functionality remains largely invariant to environmental changes within a single cell type in yeast. In stark contrast, the human enhancer analysis (Figure 6) reveals pronounced cell-type specificity in TFBS functionality. Across HepG2, K562, and SK-N-SH cell lines, we identified numerous TFBSs that exhibit different regulatory roles depending on the cellular context. For instance, certain TFBSs function as positive regulators in one cell line while acting as negative regulators or remaining neutral in others. The presence of cell line-specific TFBSs in each functional category (positive, neutral, and negative) provides compelling evidence for the context-dependent activity of transcription factors in human cells. These comparative findings highlight a fundamental difference in regulatory mechanisms: while yeast exhibits consistent TFBS functionality across different environmental conditions within the same cell type, human enhancers demonstrate significant cell-type specificity in how transcription factors influence gene expression. This cell-type dependent

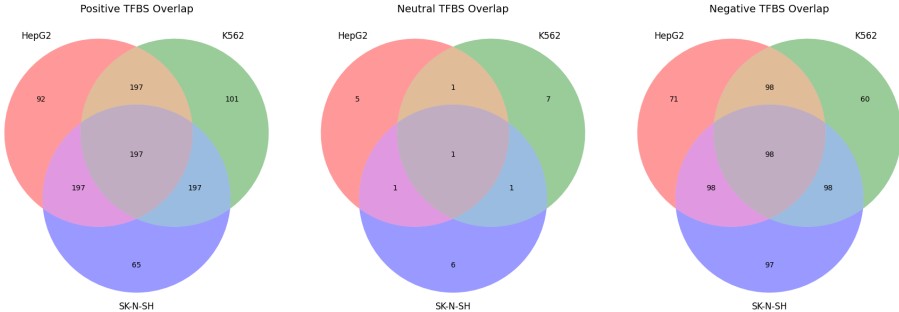

Figure 6: Venn diagrams categorizing TFBSs by their functional roles (positive, neutral, or negative) across three human cell lines (HepG2, K562, and SK-N-SH). The significant number of cell line-specific TFBSs in each category demonstrates that transcription factors can adopt distinct regulatory functions depending on cellular context, revealing the cell-type specificity of TFBS activity in human enhancers.

regulatory flexibility in humans likely contributes to the greater complexity and diversity of gene expression patterns observed across different human tissues. Our results emphasize the importance of considering the cell-type context when designing synthetic enhancers for human applications, as TFBS functionality cannot be assumed to remain constant across different cellular environments.

## E    DETAILS OF LIGHTGBM

We utilized LightGBM (Ke et al., 2017) to train models that directly predict CRE fitness based on TFBS frequency features, enabling us to infer the cell type-specific roles of individual TFBSs. To infer the regulatory impact of each TFBS, we first define the TFBS frequency feature of a sequence $X$ as a vector $\mathbf{h}(X) = [h_1(X), h_2(X), \ldots, h_n(X)]$, where $h_i(X)$ denotes the frequency of the $i$-th TFBS in sequence $X$. The LightGBM model is trained to map the TFBS frequency features to the corresponding fitness values of sequences, using the objective function:

$$\min_{\gamma} \sum_{X \in \mathcal{D}_{\text{low}}} d\left(f(X), \hat{f}(\mathbf{h}(X); \gamma)\right), \tag{7}$$

where $f(X)$ is the true fitness value of sequence $X$, $\hat{f}(\mathbf{h}(X); \gamma)$ is the fitness value predicted by the LightGBM model parameterized by $\gamma$ using the TFBS frequency feature vector $\mathbf{h}(X)$. The term $d\left(f(X), \hat{f}(\mathbf{h}(X); \gamma)\right)$ represents a distance metric measuring the discrepancy between the true and predicted fitness values.

For each dataset, we independently trained a LightGBM regression model. The specific hyperparameters used in our model are listed in Table 9 in Section 4.2.

| Parameter | Value |
|---|---|
| Objective | Regression |
| Metric | MAE |
| Boosting Type | GBDT |
| Number of Leaves | 63 |
| Learning Rate | 0.05 |
| Feature Fraction | 0.7 |
| Seed | Random State |

Table 9: Hyperparameters used for training the LightGBM regression model.

| Metric | yeast | | human | | |
|---|---|---|---|---|---|
| | complex | defined | hepg2 | k562 | sknsh |
| MAE | 0.63 | 0.65 | 0.65 | 0.65 | 0.66 |
| RMSE | 0.63 | 0.64 | 0.56 | 0.57 | 0.58 |

Table 10: Ablation study comparing different metrics on CRE fitness prediction for yeast and human datasets for training LightGBM models

We experimented with various loss functions corresponding to the distance metric $d$ in Equation equation 7, specifically testing Mean Absolute Error (MAE) and Root Mean Square Error (RMSE) as defined below:

$$d_{\mathrm{MAE}}(f(X), \hat{f}(\mathbf{h}(X); \gamma)) = \frac{1}{|\mathcal{D}_{\mathrm{low}}|} \sum_{X \in \mathcal{D}_{\mathrm{low}}} \left| f(X) - \hat{f}(\mathbf{h}(X); \gamma) \right| \tag{8}$$

$$d_{\mathrm{RMSE}}(f(X), \hat{f}(\mathbf{h}(X); \gamma)) = \sqrt{\frac{1}{|\mathcal{D}_{\mathrm{low}}|} \sum_{X \in \mathcal{D}_{\mathrm{low}}} \left( f(X) - \hat{f}(\mathbf{h}(X); \gamma) \right)^2} \tag{9}$$

We also varied other hyperparameters including learning rates {0.01, 0.05} and number of leaves {31, 63}. Our preliminary experiments indicate that learning rate and the number of leaves have minimal impact on the results, while the choice of loss function significantly affects performance. The results for these two loss functions are shown in Table 10. We found that MAE consistently outperformed RMSE. This is likely because TFBS occurrences are highly sparse, and MAE tends to perform better with sparse features (Willmott & Matsuura, 2005). Therefore, we selected MAE as the final loss function for training our LightGBM models.

## F    CHALLENGES IN EVALUATING GENERATED DNA SEQUENCES

Unlike molecules and proteins (Uehara et al., 2024), which inherently possess well-defined physical and chemical properties, DNA sequences lack such structural constraints. For example, molecular structures are subject to physical properties like bond angles and energy states, while protein sequences are evaluated based on their 3D folding stability and interactions, making it straightforward to filter out physically implausible designs. Therefore, in molecule and protein design, oracle-predicted fitness is often supplemented with physical property constraints to ensure the plausibility of generated candidates. This helps exclude a significant number of physically infeasible structures, enhancing the relevance of the optimization process. However, DNA sequences pose a unique challenge in this regard. Unlike molecules or proteins, DNA's plausibility cannot be easily assessed through physical properties, as its functional attributes are primarily determined by its interaction with transcription factors and other regulatory proteins in a context-specific manner. Furthermore, current MPRA datasets are typically generated from random sequences, meaning there is no inherent concept of plausibility in the data itself.

Our observations further highlight this challenge. In our experiments, we found that the novelty values of generated DNA sequences were disproportionately high compared to the initial low-fitness sequences, making the novelty metric less informative. This behavior suggests that DNA sequences tend to diverge significantly from their starting points during optimization, regardless of their biological relevance or plausibility. Due to these limitations, we exclude the *Novelty* metric and instead focus on evaluating the generated sequences using *Fitness* and *Diversity* metrics, which better capture the optimization objectives for CRE design.

However, we believe that developing appropriate metrics for evaluating DNA sequence plausibility remains highly significant for CRE design. Beyond essential experimental validation (Gosai et al., 2024; Vaishnav et al., 2022), future work could incorporate assessment methods proposed in Zeng et al. (2024) and Avdeyev et al. (2023) to more comprehensively evaluate the biological plausibility of generated sequences.

# G   MORE DETAILS OF TACO

## G.1   ALGORITHM OVERVIEW

The overview of our algorithm TACO (the setting is in Section 4.2)is shown in Algorithm 1.

---

**Algorithm 1** TACO: RL-Based Fine-tuning for Autoregressive DNA Models

---

**Require:** Low-fitness dataset $\mathcal{D}_{\text{low}}$, TFBS vocabulary $\mathcal{T}$, Oracle $q_\psi$, Pretrained AR model $\pi_\theta$, Number of Optimization Rounds $E$
 1: **Preprocessing:**
 2: Train LightGBM model on TFBS frequency features $\mathbf{h}(X)$ from dataset $\mathcal{D}_{\text{low}}$
 3: Compute SHAP values $\phi_j(X)$ for each TFBS $t_j$
 4: Update TFBS rewards $r_{\text{TFBS}}(t)$ based on  equation 5
 5: **for** round $e = 1$ to $E$ **do**
 6:     Sample a batch of sequences $\{X_k\}$ (K = 256) from policy $\pi_\theta$
 7:     **for** each sequence $X_k$ **do**
 8:         **for** time step $i = 1$ to $L$ **do**
 9:             Generate nucleotide $a_i$ using $\pi_\theta(a_i|a_{<i})$
10:             Observe state $s_i = (a_1, \ldots, a_i)$
11:             **if** $a_i$ results in TFBS $t \in \mathcal{T}$ **then**
12:                 Assign reward $r(s_{i-1}, a_i) \leftarrow r_{\text{TFBS}}(t)$
13:             **else**
14:                 Assign reward $r(s_{i-1}, a_i) \leftarrow 0$
15:             **end if**
16:         **end for**
17:         Obtain fitness reward $r_{\text{fitness}}$ from oracle $q_\psi(X_k)$
18:         Compute total reward $R \leftarrow \sum_{i=1}^{L} r(s_{i-1}, a_i) + r_{\text{fitness}}$
19:     **end for**
20:     Update policy $\pi_\theta$ using REINFORCE:
$$\theta \leftarrow \theta + \alpha \nabla_\theta \mathbb{E}_{\pi_\theta} [R \log \pi_\theta(a_i|s_{i-1})]$$
21: **end for**

---

## G.2   THE EFFECT OF AUXILIARY RL DESIGNS

As illustrated in Figure 7, we systematically evaluate two critical components of our auxiliary designs presented in Section 3.2: the hill-climb replay buffer and entropy regularization. First, we investigate the impact of the hill-climb replay buffer, which preserves past experiences with high fitness values. Our results demonstrate that incorporating this replay buffer significantly enhances the maximum fitness values achieved during exploration, aligning with findings from previous research (Lee et al., 2024; Ghugare et al., 2024). Subsequently, we examine the effectiveness of entropy regularization, which is specifically designed to promote exploration by increasing policy randomness and preventing premature convergence to suboptimal solutions. Our experimental results reveal that this approach successfully leads to improved action diversity, underscoring its value in facilitating a more comprehensive exploration of the solution space.

# H   OFFLINE MODEL-BASED OPTIMIZATION

In Section 4.2, we present results under an active learning setting (Lee et al., 2024), which assumes easy access to a perfect oracle for evaluating generated CRE sequences in the optimization process. However, this setting can lead to optimization processes that overfit to an imperfect oracle (trained only with observed data).

Here, we consider an alternative offline model-based optimization(MBO) setting (Reddy et al., 2024; Uehara et al., 2024), which assumes that accessing the true oracle is costly, but some labeled offline data is available. In this setting, a surrogate model is trained on the offline dataset to guide the

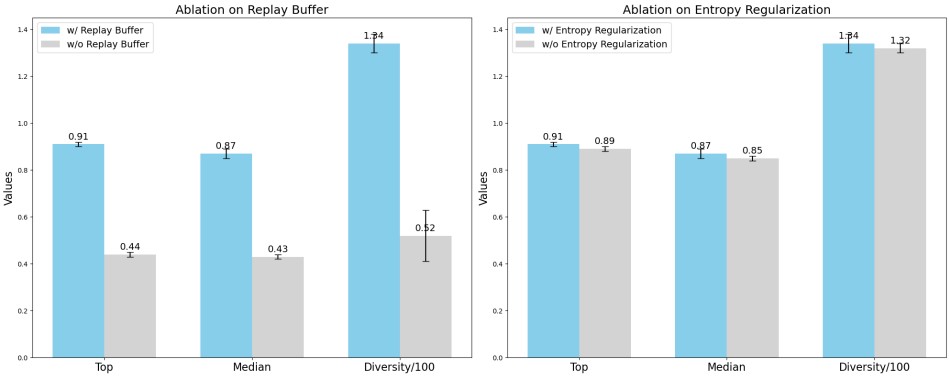

Figure 7: Ablation study on supporting RL designs.

optimization process, and the final sequences are evaluated by the oracle. This approach helps mitigate overfitting to a "man-made oracle" trained on limited data.

Figure 8 illustrates an example (a single run on the yeast complex dataset). The left panel shows the curve of the Top fitness predicted by the surrogate as the iterations progress. Since the optimization is guided by the surrogate, the curve continues to increase. Initially, the right panel (representing the Top fitness as predicted by the oracle) also rises steadily. However, around iteration 80, there is a sharp increase in the surrogate's predicted fitness, while the oracle's predicted fitness exhibits a brief spike before declining. This behavior suggests that at 80 iterations, the optimization process discovers a seemingly high-fitness point. However, the surrogate, believing this direction to be correct, continues optimizing, leading to an overestimation of the fitness. The oracle's actual score, however, does not continue to increase significantly. This example demonstrates that in real optimization processes, the surrogate can be misled by spurious data points, further emphasizing the importance of the offline MBO setting.

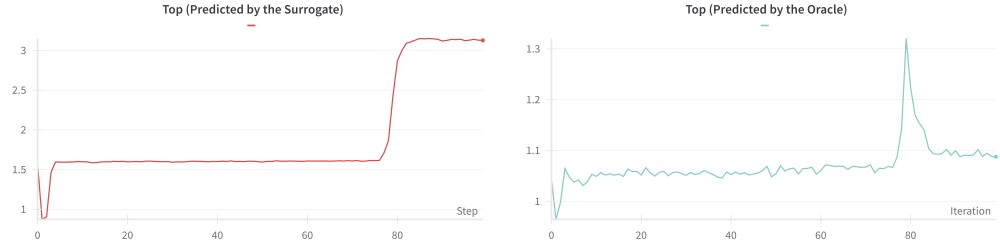

Figure 8: Left: The curve of Top fitness predicted by the surrogate during iterations. Right: The corresponding Top fitness predicted by the oracle. The discrepancy highlights the potential for the surrogate to overestimate fitness due to spurious data points, emphasizing the need for offline MBO settings.

Specifically, we still use the oracle for the final evaluation of sequences, but we sub-sample a portion of the data to create a predefined offline dataset. Following Uehara et al. (2024), the sub-sampling strategy involves randomly splitting the dataset in half and selecting sequences with fitness values below the 95th percentile to simulate a real-world scenario where observed data may have a lower ceiling. This dataset is referred to as $D_{\text{offline}}$. A surrogate model is trained on $D_{\text{offline}}$, and the optimization process proceeds similarly to Section 4.2, except that each iteration is guided by the surrogate, with the oracle used only for final quality evaluation of the generated sequences.

# I   DISCUSSION AND IMPLEMENTATION DETAILS OF CONDITIONAL GENERATIVE MODELS

Although the objectives of generative models and optimization methods differ, both aim to propose samples that deviate from the observed real-world data. To this end, we include a discussion and comparison with SOTA generative models.

Let the data distribution be denoted as $P(x)$, where each data point $x$ is paired with a label $y$ (e.g., the fitness of a CRE). The full dataset observed in the real world is represented as $D = \{(x_i, y_i)\}_{i=1}^{N}$. In biological sequence data, $x$ typically follows a reasonable underlying distribution $P(x)$, which can be approximated using a generative model $P_{\text{pre}}(x)$ without requiring knowledge of $y$. However, directly sampling from $P_{\text{pre}}(x)$ often yields sequences with low fitness, as the distribution of $y$ values (e.g., high-fitness regions) is typically narrow and sparsely represented in the data. Thus, an unconditional generative model is generally ineffective for designing biological sequences.

To address this limitation, conditional generative modeling can be employed. By training a model to approximate $P(x \mid y)$ using the offline labeled dataset $D_{\text{offline}}$, we can condition on high observed fitness values $y$ to theoretically generate high-fitness sequences. Formally, given a dataset where $y$ is partitioned into discrete bins or ranges (e.g., high-fitness values), the conditional generative model is trained to maximize the likelihood. Subsequently, sequences are generated by sampling $x$ conditioned on $y$ values corresponding to high fitness. However, in practice, this approach often underperforms because the distribution of high $y$ values is extremely narrow, and the model struggles to accurately capture this region.

We compare our method against recent generative models, including the autoregressive generative model reglm (Lal et al., 2024) and the discrete diffusion model DDSM (Avdeyev et al., 2023). For evaluations, we adopted conditional generation strategies for both models. Specifically: **regLM**: The official pretrained weights were used. Sequences were generated by conditioning on the prefix label corresponding to the highest fitness score in each dataset. **DDSM**: This model was trained on our offline dataset, where labels above the 95th percentile were set to $y = 1$, and the remaining labels were set to $y = 0$ following Uehara et al. (2024). The conditional diffusion model was then trained using this binary labeling scheme, and sequences were generated by conditioning on $y = 1$ for evaluation.

As shown in Table 18, both regLM and DDSM exhibit high diversity in their generated sequences but fail to match the fitness values achieved by optimization-based methods. This limitation arises because generative models are designed to fit the observed data distribution $P(x \mid y)$, and as such, their generated sequences are inherently constrained by the data's fitness distribution. It is also worth noting that reglm utilized official pretrained weights, which may have been exposed to data with higher fitness scores than our offline dataset. Even with this advantage, it fails to outperform optimization-based methods. In contrast, our method builds upon a pretrained distribution $P_{\text{pre}}(x)$ and further proposes new sequences by iteratively optimizing $P_{\text{pre}}(x)$ through feedback from an oracle or surrogate. The ultimate goal is to reshape the distribution so that high-fitness sequences become more accessible during sampling.

# J   ANALYSIS OF GRADIENT ASCENT'S PERFORMANCE IN OFFLINE MBO SETTINGS.

As stated in Reddy et al. (2024), in offline MBO settings, directly applying Gradient Ascent (GAs) to a surrogate is expected to generate adversarial examples with poor performance. However, in our experiments, we did not observe this phenomenon. Instead, performing GAs directly led to surprisingly good results, which is an unexpected finding. To the best of our knowledge, prior CRE design work has not extensively explored GAs, with the exception of Reddy et al. (2024). However, this study did not include an ablation study on regularization terms. Therefore, in the context of machine learning-based DNA CRE design, it remains an open question whether directly applying GAs to a differentiable surrogate could generate adversarial examples with poor performance. We acknowledge that, in an ideal scenario, surrogates trained on local data may indeed generate adversarial examples, and we believe this issue warrants further attention in future research.

To further address this concern, we validated GAs performance across different fitness quantiles (95 shown in Figure 9a, 80 shown in Figure 9b, 60 shown in Figure 9c) using K562 cells (our default offline MBO setting was the 95th percentile). Our GAs implementation directly operates on the one-hot encoded probability simplex following Reddy et al. (2024), which allows for smooth updates during optimization. Therefore, we report both the results on the one-hot-encoded simplex (*Prob*) and the hard-decoded sequences optimized (*Sequence*) in each iteration. We reported the scores predicted by both the surrogate and oracle for these two representations. Our findings indicate: (1) For the 95th percentile, as shown in Figure 9a, the fitness in the sequence space initially rises sharply but then drops. For the 60th percentile, as shown in Figure 9c, a similar pattern is observed in the oracle's predictions within the Prob space. This reveals **a gap between the surrogate and the oracle**, as the surrogate's predictions consistently increase. This aligns with our expectations of the offline MBO setting, i.e., the surrogate cannot fully reflect the oracle. (2) However, the oracle's predictions do show significant improvement at the start, indicating that directly applying GAs to the surrogate can still benefit the oracle's results. This suggests that, under the current CRE data partitioning strategy, even a surrogate trained on low-fitness subsets can reasonably capture the trends of the oracle's predictions (although the surrogate itself, having never encountered high-fitness data, predicts much lower upper bounds).

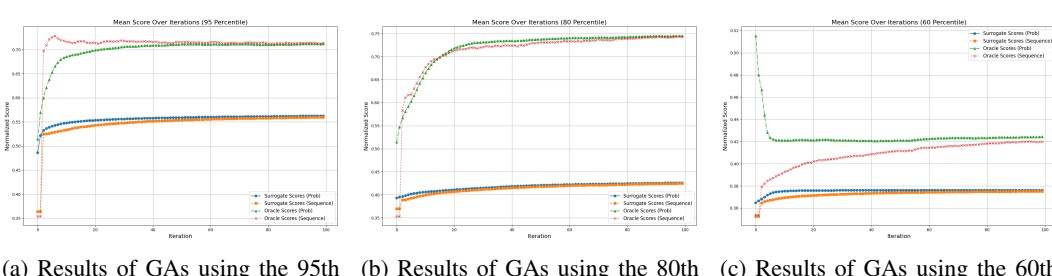

(a) Results of GAs using the 95th percentile subset.

(b) Results of GAs using the 80th percentile subset.

(c) Results of GAs using the 60th percentile subset.

Figure 9: Comparison of GA results using different percentile subsets (95th, 80th, and 60th).

Based on the current evidence, we believe that the observed good performance of GAs may be an inherent property (possibly related to the inherent data distribution of CREs and our current data partitioning strategy). Even a surrogate trained on the 60th percentile subset can achieve decent performance. This is an interesting question for future research in CRE design.

However, we emphasize that our primary focus is on designing optimization algorithms rather than relying on **a differentiable surrogate**. Our current offline MBO setting has already made the task more challenging, achieving the intended goal of designing an offline MBO setting. Nevertheless, we do not yet fully understand why GAs does not lead to significantly poor results. Figuring this out is left for future work, but we believe it is crucial for machine-learning-driven CRE design.

Besidies, we have also added the results of different methods (including GAs) guided by a surrogate trained on the 60th percentile shown in Table 11, Table 12 and Table 13. It can be observed that, despite the significant gap between the surrogate and the oracle under the 60th percentile training, GAs still achieves relatively good performance. Notably, under the 60th percentile setting, PEX, which performes well at the 95th percentile, shows moderate results, while CMAES, which previously performes the worst, achieves excellent performance. Our TACO, in this setting, continues to maintain SOTA results.

| Model | Top ↑ | Medium ↑ | Diversity ↑ | Emb Similarity ↓ |
|---|---|---|---|---|
| PEX | 0.54 (0.02) | 0.48 (0.02) | 16.4 (5.13) | 0.80 (0.03) |
| AdaLead | 0.50 (0.05) | 0.42 (0.02) | **146.6** (2.61) | **0.36** (0.03) |
| BO | 0.46 (0.04) | 0.41 (0.02) | 41.2 (6.91) | 0.71 (0.07) |
| CMAES | 0.55 (0.04) | 0.41 (0.02) | 78.4 (3.97) | 0.41 (0.05) |
| GAs | 0.59 (0.01) | 0.51 (0.01) | 136.20 (0.40) | 0.80 (0.01) |
| TACO | **0.56** (0.08) | **0.50** (0.08) | 134.6 (14.03) | 0.77 (0.08) |

Table 11: Results of HepG2 (60 Percentile).

| Model | Top ↑ | Medium ↑ | Diversity ↑ | Emb Similarity ↓ |
|---|---|---|---|---|
| PEX | 0.50 (0.06) | 0.45 (0.04) | 13.6 (2.19) | 0.87 (0.02) |
| AdaLead | 0.62 (0.09) | **0.49** (0.10) | **138.6** (20.7) | **0.55** (0.11) |
| BO | 0.55 (0.03) | 0.44 (0.03) | 43.6 (7.37) | 0.66 (0.11) |
| CMAES | **0.66** (0.07) | 0.47 (0.05) | 79.0 (4.36) | 0.49 (0.06) |
| GAs | 0.52 (0.02) | 0.43 (0.00) | 126.60 (1.20) | 0.84 (0.01) |
| TACO | 0.63 (0.04) | 0.48 (0.02) | 132.4 (25.7) | 0.62 (0.21) |

Table 12: Results of K562 (60 Percentile).

| Model | Top ↑ | Medium ↑ | Diversity ↑ | Emb Similarity ↓ |
|---|---|---|---|---|
| PEX | 0.50 (0.11) | 0.47 (0.11) | 19.6 (3.21) | 0.74 (0.04) |
| AdaLead | 0.55 (0.06) | 0.44 (0.03) | **141.6** (13.24) | 0.54 (0.09) |
| BO | 0.55 (0.09) | 0.46 (0.05) | 48.8 (11.65) | 0.72 (0.09) |
| CMAES | 0.61 (0.09) | 0.44 (0.03) | 75.2 (1.92) | **0.50** (0.05) |
| GAs | 0.48 (0.01) | 0.42 (0.00) | 140.00 (0.63) | 0.59 (0.01) |
| TACO | **0.69** (0.04) | **0.57** (0.06) | 135.6 (5.5) | 0.78 (0.06) |

Table 13: Results of S-KN-SH (60 Percentile).

## K    MORE EXPERIMENTAL RESUTLS

Since many conclusions are consistent across different datasets and settings,we include a significant portion of the experimental results here in the Apeendix. The complete experimental results for yeast under the oracle-guided optimization setting (Section 4.2) are presented in Figure 14. The results for offline MBO (Section 4.3) are detailed in Table 15, Table 16, Table 17, Table 18, and Table 19.

| | Yeast Promoter (Complex) | | | | | | | | |
|---|---|---|---|---|---|---|---|---|---|
| **Method** | easy | | | middle | | | hard | | |
| | Top ↑ | Medium ↑ | Diversity ↑ | Top ↑ | Medium ↑ | Diversity ↑ | Top ↑ | Medium ↑ | Diversity ↑ |
| PEX | 1 | 1 | 8.6 (1.14) | 1 | 1 | 8.4 (1.95) | 1 | 1 | 9.8 (1.48) |
| AdaLead | 1 | 1 | 8.8 (1.3) | 1 | 1 | 9.0 (1.58) | 1 | 1 | 7.6 (0.89) |
| BO | 1 | 1 | 23.4 (1.52) | 1 | 1 | 22.6 (1.34) | 1 | 1 | 25.0 (5.57) |
| CMAES | 1 | 0.78 (0.13) | 30.2 (2.68) | 1 | 0.85 (0.02) | 29.4 (1.52) | 1 | 0.79 (0.09) | 30.0 (2.5) |
| DNARL | 1 | 1 | 8.6 (2.14) | 1 | 1 | 10.2 (1.14) | 1 | 1 | 7.7 (0.48) |
| TACO | 1 | 1 | **52.2** (1.92) | 1 | 1 | **48.8** (5.36) | 1 | 1 | **52.8** (2.77) |

| | Yeast Promoter (Defined) | | | | | | | | |
|---|---|---|---|---|---|---|---|---|---|
| **Method** | easy | | | middle | | | hard | | |
| | Top ↑ | Medium ↑ | Diversity ↑ | Top ↑ | Medium ↑ | Diversity ↑ | Top ↑ | Medium ↑ | Diversity ↑ |
| PEX | 1 | 1 | 9.2 (0.84) | 1 | 1 | 9.2 (1.79) | 1 | 1 | 9.8 (2.59) |
| AdaLead | 1 | 1 | 8.0 (2.35) | 1 | 1 | 7.0 (1.0) | 1 | 1 | 6.4 (0.55) |
| BO | 1 | 1 | 23.0 (1.58) | 1 | 1 | 22.8 (2.28) | 1 | 1 | 23.0 (1.87) |
| CMAES | 1 | 0.26 (0.36) | 30.0 (2.92) | 1 | 0.48 (0.17) | 29.8 (1.3) | 1 | 0.44 (0.33) | 30.4 (2.3) |
| DNARL | 1 | 1 | 11.6 (3.04) | 1 | 1 | 18.5 (3.0) | 1 | 1 | 10.2 (1.14) |
| TACO | 1 | 1 | **43.2** (2.77) | 1 | 1 | **47.0** (4.64) | 1 | 1 | **49.6** (3.65) |

Table 14: Performance comparison on yeast promoter datasets (Guided by the Oracle).

| Model | Top ↑ | Medium ↑ | Diversity ↑ | Emb Similarity ↓ |
|---|---|---|---|---|
| PEX | 1.16 (0.09) | 1.12 (0.08) | 11.4 (57.60) | 0.98 (0.01) |
| AdaLead | 1.06 (0.02) | 1.00 (0.02) | 57.6 (0.55) | 0.95 (0.00) |
| BO | 1.09 (0.02) | 1.03 (0.03) | 24.4 (4.77) | 0.97 (0.01) |
| CMAES | 1.06 (0.07) | 0.70 (0.12) | 29.20 (0.45) | 0.75 (0.05) |
| regLM | 1.02 (0.00) | 0.94 (0.00) | 59.00 (0.00) | 0.91 (0.01) |
| ddsm | 0.94 (0.02) | 0.79 (0.01) | 58.20 (0.40) | 0.81 (0.01) |
| TACO | 1.06 (0.01) | 0.98 (0.01) | 57.4 (1.34) | 0.93 (0.01) |

Table 15: Offline MBO (95 Percentile) results (yeast promoter, complex).

| Model | Top ↑ | Medium ↑ | Diversity ↑ | Emb Similarity ↓ |
|---|---|---|---|---|
| PEX | 1.19 (0.15) | 1.10 (0.16) | 10.40 (2.61) | 0.98 (0.01) |
| AdaLead | 1.02 (0.04) | 0.98 (0.04) | 8.20 (1.79) | 0.98 (0.01) |
| BO | 1.06 (0.03) | 1.02 (0.02) | 26.00 (2.24) | 0.97 (0.01) |
| CMAES | 0.79 (0.10) | 0.39 (0.12) | 30.80 (2.05) | 0.59 (0.05) |
| regLM | 0.98 (0.01) | 0.89 (0.01) | 58.80 (0.40) | 0.90 (0.00) |
| DDSM | 0.92 (0.02) | 0.81 (0.00) | 56.20 (0.40) | 0.86 (0.01) |
| TACO | 1.10 (0.05) | 1.03 (0.04) | 46.00 (1.87) | 0.97 (0.01) |

Table 16: Offline MBO (95 Percentile) results (yeast promoter, defined).

| Model | Top ↑ | Medium ↑ | Diversity ↑ | Emb Similarity ↓ |
|---|---|---|---|---|
| PEX | 0.75 (0.01) | 0.73 (0.01) | 13.6 (4.51) | 0.98 (0.01) |
| AdaLead | 0.59 (0.01) | 0.52 (0.04) | 34.2 (59.15) | 0.84 (0.16) |
| BO | 0.65 (0.09) | 0.61 (0.10) | 40.2 (6.14) | 0.83 (0.13) |
| CMAES | 0.57 (0.03) | 0.41 (0.03) | 77.2 (2.28) | 0.45 (0.04) |
| regLM | 0.65 (0.01) | 0.48 (0.02) | 150.00 (0.00) | 0.28 (0.02) |
| DDSM | 0.41 (0.00) | 0.41 (0.00) | 15.40 (0.49) | 0.99 (0.00) |
| TACO | 0.69 (0.03) | 0.60 (0.05) | 141.2 (1.92) | 0.82 (0.05) |

Table 17: Offline MBO (95 Percentile) results (human enhancer, HepG2).

| Model | Top ↑ | Medium ↑ | Diversity ↑ | Emb Similarity ↓ |
|---|---|---|---|---|
| PEX | 0.76 (0.02) | 0.73 (0.02) | 15.8 (4.97) | 0.97 (0.01) |
| AdaLead | 0.66 (0.08) | 0.58 (0.06) | 63.2 (70.01) | 0.88 (0.12) |
| BO | 0.71 (0.07) | 0.64 (0.08) | 43.6 (6.91) | 0.87 (0.04) |
| CMAES | 0.66 (0.02) | 0.44 (0.03) | 79.2 (3.83) | 0.35 (0.03) |
| regLM | 0.69 (0.02) | 0.47 (0.01) | 149.60 (0.49) | 0.38 (0.02) |
| DDSM | 0.43 (0.00) | 0.40 (0.00) | 93.40 (0.49) | 0.80 (0.00) |
| TACO | 0.75 (0.09) | 0.72 (0.10) | 102.6 (20.14) | 0.97 (0.04) |

Table 18: Offline MBO (95 Percentile) results (human enhancer, K562).

| Model | Top ↑ | Medium ↑ | Diversity ↑ | Emb Similarity ↓ |
|---|---|---|---|---|
| PEX | 0.69 (0.01) | 0.68 (0.00) | 17.8 (3.90) | 0.98 (0.01) |
| AdaLead | 0.59 (0.08) | 0.56 (0.08) | 8.6 (2.30) | 0.96 (0.03) |
| BO | 0.61 (0.09) | 0.52 (0.08) | 42.4 (4.77) | 0.80 (0.08) |
| CMAES | 0.58 (0.05) | 0.42 (0.03) | 78.6 (1.14) | 0.40 (0.06) |
| regLM | 0.61 (0.00) | 0.47 (0.01) | 149.60 (0.49) | 0.38 (0.03) |
| DDSM | 0.54 (0.00) | 0.49 (0.00) | 102.20 (1.17) | 0.91 (0.01) |
| TACO | 0.68 (0.08) | 0.62 (0.08) | 121.4 (7.86) | 0.90 (0.03) |

Table 19: Offline MBO (95 Percentile) results (human enhancer, S-KN-SH).

