# OpenReview forum: "Regulatory DNA Sequence Design with Reinforcement Learning"
_ICLR.cc/2025/Conference — ICLR 2025 Poster_

### Official Review · Reviewer_BjdH · 2024-11-01

**Soundness:** 2
**Presentation:** 3
**Contribution:** 2
**Rating:** 5
**Confidence:** 4

**Summary:**

The authors propose a reinforcement learning (RL) method for designing cis-regulatory elements. The method uses a pre-trained autoregressive model of DNA sequences as its policy and fine-tunes this model throughout the training process. The method incorporates domain knowledge by adding a reward that encourages generated sequences to contain known transcription factor binding site motifs. The authors test their method against a number of relevant baselines on a set of yeast promoter and human enhancer design tasks, showing improved performance over the baselines.

**Strengths:**

* The paper is clearly written. The problem and method are well-motivated and clearly explained.

**Weaknesses:**

* The evaluation tasks in this paper do not represent a realistic scenario for biological sequence design and do not follow established practices from the literature. Biological sequence design with machine learning is best characterized as an offline model-based optimization (MBO) problem, where there exists a fixed dataset associating sequences to fitness, but intermediate fitness measurements can not be collected during the process of optimization. This is because biological experiments are expensive but highly parallelizable, so it it usually only economical to collect a large number of measurements at once, rather than a small number of intermediate measurements. A possible approach to solving this offline MBO problem is train a model to predict fitness from sequence, and then use this "surrogate" model to guide an optimization procedure (which could be RL or another process). Note that the surrogate model is sometimes misleadingly referred to as an oracle. The most important concern with these methods is that the surrogate model is not a good model of the ground truth fitness function in regions that are out-of-distribution (OOD) of the fixed training set. Thus, the major methodological advancements in offline MBO have been aimed at controlling the optimization such that it does not produce sequences that are OOD of the training set. This problem is discussed extensively in, e.g., "Conditioning by adaptive sampling"  (2019) by Brookes et al.,
 and "Conservative Objective Models for Effective Offline Model-Based Optimization" (2021) by Trabucco et al. In the paper under review, the authors also train a surrogate/oracle model to guide their optimization but make two choices that minimize the practical relevance of their evaluation tasks:
    1. They train the oracle on the complete dataset, even though the intention of the task is to design high fitness sequences using only low fitness data. Therefore, when they query the oracle to calculate rewards, they are treating the oracle as a ground truth fitness function that they are collecting intermediate rewards for. As discussed above, this is unrealistic for biological sequence design.
    2. They use predictions from the oracle as the final evaluation criteria. This again treats the oracle as the ground truth fitness function.
The combination of choices (1) and (2) mean that the evaluation tasks simply measure the ability of the method to optimize the oracle. As discussed in the citations above, optimizing the oracle without controlling for OOD concerns will usually result in designing unrealistic sequences that are OOD of the initial training set. Evaluation of methods for biological sequence require careful consideration of these factors, which has led to the introduction of evaluation frameworks such as FLEXS (referenced in the paper); these factors must be taken into account when designing tasks to evaluate the TACO method introduced in this paper.
* A recent method has been introduced that uses Conservation Model Objectives (COMs) to design CREs (["Designing Cell-Type-Specific Promoter Sequences Using Conservative Model-Based Optimization" (2024) by Reddy, et. al.](https://www.biorxiv.org/content/10.1101/2024.06.23.600232v1)). This method considers the factors discussed in the previous bullet, and contains strong validation with in-vitro experiments. It is not cited in the paper under review and not compared against. In order to be a strong contribution, the paper under review must cite this COMs paper and compare against the method.
* The novelty of the method seems overstated as written. In particular, RL has been applied to biological sequence design in Angermueller et. al, 2019, which is not made clear in the Related Work section. Given this, the novelty of the authors' method is in the use of an autoregressive model as the policy, and the addition of the TFBS reward. This should be made more clear by making clear the contributions of Angermueller et. al, 2019 and how the authors' method improves over this.
* As mentioned above, the authors introduce two novel concepts to the use of RL in sequence design. In order to understand the impact of these concepts, they should be fully ablated on all of the evaluation tasks. In particular, it would be informative to know how a different policy model performs with the TFBS reward and how the autoregressive policy performs without the TFBS rewards on all of the design tasks. This will clarify the key contributions of the paper.
* Figures 3A and 5A are below acceptable quality for a publication at ICLR. Both appear to be screenshots and contain either unreadable or poorly labeled axes/subplots.
* The References section contains incorrect and inconsistent citation formatting. In particular, many titles are lower case when they should be upper case and inconsistent information is included in the citation (e.g. sometimes URLs are provided and other times they are not). Also, many citations refer to an arXiv pre-print, rather than the published version of a paper; this must be checked and fixed.

Minor errors:
* Second sentence of abstract is grammatically incorrect (should "CRE" be plural?)
* Equation 1: p_theta should be pi_theta
* Line 368: (Hansen) is an incomplete reference

**Questions:**

* The intermediate TFBS reward only gives a reward to sequences that contain known TFBS motifs. This seems like it limits the model to recombining known motifs, rather than learning to design new motifs. Since TFBS motif design is a task that is of interest to the community (as is mentioned in Related Work), the form of the reward may limit the practical applicability of the method. Can the author's clarify whether their method is able to design novel TFBS motifs? If not, can the author's clarify whether this is an important limitation or not?

---

> ### Author Response · Authors · 2024-11-25
> **Response to Reviewer BjdH (1)**
>
> Thanks for your detailed and insightful comments. We will address each of your concerns point by point in the following response.
>
> **Q1**: The evaluation tasks in this paper do not represent a realistic scenario for biological sequence design and do not follow established practices from the literature.
>
> **A1**: Thank you for this suggestion. In our revision, we have added experimental evaluations under the offline MBO framework. In this setting, we assume that only a subset of each offline dataset is available for training the surrogate model, with sequences in this subset exhibiting relatively low activities. The detailed experimental setup is provided in Section 4.3. Under this setting, the performance of all methods decreases significantly. However, the relative rankings among the models remain consistent with previous observations. As shown in Table 1, our TACO achieves the highest Diversity while maintaining fitness optimization comparable to the best-performing methods. Results on additional datasets are provided in Appendix M of the revised draft.
>
> **Table 1** Offline MBO results for human enhancers (K562).
> | **Model**   | **Top ↑**  | **Medium ↑**  | **Diversity ↑**  | **Emb Similarity ↓** |
> |-------------|------------|---------------|------------------|----------------------|
> | **PEX**     | **0.76 (0.02)** | **0.73 (0.02)** | 15.8 (4.97)     | 0.97 (0.01)         |
> | **AdaLead** | 0.66 (0.08) | 0.58 (0.06)    | 63.2 (70.01)     | 0.88 (0.12)         |
> | **BO**      | 0.71 (0.07) | 0.64 (0.08)    | 43.6 (6.91)      | 0.87 (0.04)         |
> | **CMAES**   | 0.66 (0.02) | 0.44 (0.03)    | 79.2 (3.83)      | **0.35 (0.03)**     |
> | **regLM**   | 0.69 (0.02) | 0.47 (0.01)    | **149.60 (0.49)**| 0.38 (0.02)     |
> | **DDSM**    | 0.43 (0.00) | 0.40 (0.00)    | 93.40 (0.49)     | 0.80 (0.00)         |
> | **TACO**    | 0.75 (0.09) | 0.72 (0.10)    | 102.6 (20.14)| 0.97 (0.04)         |

---

> ### Author Response · Authors · 2024-11-25
> **Response to Reviewer BjdH (2)**
>
> **Q2**: A recent method has been introduced that uses Conservation Model Objectives (COMs) to design CREs. It is not cited in the paper under review and not compared against. In order to be a strong contribution, the paper under review must cite this COMs paper and compare against the method.
>
> **A2**: We highly appreciate the contributions made by [1] in the field of promoter design. In the revision, we have discussed this work in Section 2. However, we believe the focus of [1] differs slightly from our approach. Specifically, this work emphasizes critical aspects of the practical design pipeline, such as defining the Difference of Expression (DE) to optimize cell-type-specific promoters and selecting sequences with both high fitness and high discriminative power for experimental validation. We acknowledge that [1] holds significant potential in real-world CRE design scenarios.
>
> That said, [1] requires a specialized differentiable conservation-penalized surrogate and involves training models with varying penalization coefficients, among other complex techniques. In contrast, our work focuses on optimization algorithms and does not rely on training sophisticated or differentiable surrogates. Our benchmarks primarily evaluate optimization algorithms under the guidance of a uniformly trained, unmodified surrogate model.
>
> In summary, there are several reasons why a direct comparison was not feasible:
> (1) The datasets used in [1] differ from those employed in our study.
> (2) The surrogate training methodologies differ, while our approach does not depend on specific surrogate training strategies or differentiability requirements.
> (3) The preprint version of [1] does not provide source code, making direct comparison impractical.
>
> We hope these points clarify the distinctions between our work and [1], as well as the challenges associated with performing a direct comparison.
>
> To further address the reviewers' concerns, we quickly implemented a Gradient Ascent (GAs) algorithm based on the shared surrogate used across all baselines within the limited rebuttal timeframe. The details of the Gradient Ascent implementation were entirely derived from the methods described in [1]. As shown in Table 2, we found that directly performing Gradient Ascent on the surrogate achieves surprisingly strong results. Specifically, the Top metric surpasses TACO by 0.01, while Medium is lower by 0.07. However, Diversity is far better than TACO. These results are surprising because the samples generated by GAs are classified as adversarial samples in [1], meaning their optimization direction should theoretically conflict with the target task. This makes the results overall quite interesting, and we believe further comparison with [1] represents an exciting direction for future work. However, it is important to emphasize that our current work does not require a differentiable surrogate, and as such, we do not prioritize comparisons with methods that rely on gradient ascent-based optimization.
>
>
> **Table 2** Comparision with Gradient Ascent (GAs) for human enhancers (K562).
> | **Model**   | **Top ↑**  | **Medium ↑**  | **Diversity ↑**  | **Emb Similarity ↓** |
> |---------|-----------|-----------|----------------|----------------|
> | PEX     | **0.76**(0.02) | **0.73**(0.02) | 15.8(4.97)     | 0.97(0.01)     |
> | AdaLead | 0.66(0.08) | 0.58(0.06) | 63.2(70.01)    | 0.88(0.12)     |
> | BO      | 0.71(0.07) | 0.64(0.08) | 43.6(6.91)     | 0.87(0.04)     |
> | CMAES   | 0.66(0.02) | 0.44(0.03) | 79.2(3.83)     | **0.35**(0.03)     |
> | reglm   | 0.69(0.02) | 0.47(0.01) | **149.60**(0.49)   | 0.38(0.02)     |
> | ddsm    | 0.43(0.00) | 0.40(0.00) | 93.40(0.49)    | 0.80(0.00)     |
> | TACO    | 0.75(0.09) | 0.72(0.1)  | 102.6(20.14)   | 0.97(0.04)     |
> | GAs     | **0.76**(0.01) | 0.65(0.01) | 146.00(0.0)    | 0.75(0.01)     |

---

> ### Author Response · Authors · 2024-11-25
> **Response to Reviewer BjdH (3)**
>
> **Q3**: The novelty of the method seems overstated as written. In particular, RL has been applied to biological sequence design in Angermueller et. al, 2019, which is not made clear in the Related Work section. Given this, the novelty of the authors' method is in the use of an autoregressive model as the policy, and the addition of the TFBS reward. This should be made more clear by making clear the contributions of Angermueller et. al, 2019 and how the authors' method improves over this.
>
> **A3**: Thank you for pointing this out.
>
> (1) First, we do cite the work of DyNA-PPO, Angermueller et al. (2019) [2], in the Related Work section. We do not intend to claim to be the first to use an autoregressive policy for DNA optimization (as DyNA-PPO also uses an autoregressive policy). Instead, we have consistently emphasized (particularly in the Method section) that one of our key contributions is introducing RL to fine-tune a pretrained autoregressive model for DNA design. We acknowledge that some statements in the Introduction section may have been unclear. We have revised the text in the revision to better emphasize our contributions.
>
> (2) DyNA-PPO focuses on DNA tasks involving the design of transcription factor binding sites (TFBSs), which are typically less than 10 base pairs in length. In contrast, our work is specifically focused on designing cis-regulatory elements (CREs). CREs are generally longer than TFBSs, and while TFBSs are often located within CREs, these two types of gene sequences have distinct effects and characteristics. As a result, applying RL to CRE design requires different policy models and reward structures. While [2] primarily focuses on improving the PPO algorithm, our approach leverages a pretrained autoregressive model as the policy to capture the underlying patterns and rules of CREs, enabling the generation of feasible candidates more efficiently. Additionally, we introduce the TFBS reward to encourage the generated CREs to contain biologically meaningful patterns informed by prior knowledge. As a task-specific model for CRE design, the results in Section 4.4 of the revision demonstrate the effectiveness of our key contributions.
>
> Thanks to your suggestion, we have revised Sections 1 and 2 to better emphasize our core contributions, explicitly clarifying that we are not the first to introduce an autoregressive policy. These advancements are now highlighted in the revised version to better situate our contributions within the existing literature.

---

> ### Author Response · Authors · 2024-11-25
> **Response to Reviewer BjdH (4)**
>
> **Q4**: As mentioned above, the authors introduce two novel concepts to the use of RL in sequence design. In order to understand the impact of these concepts, they should be fully ablated on all of the evaluation tasks. In particular, it would be informative to know how a different policy model performs with the TFBS reward and how the autoregressive policy performs without the TFBS rewards on all of the design tasks. This will clarify the key contributions of the paper.
>
> **A4**:  Thank you for this suggestion. Based on the following considerations: (1) the offline MBO setting is more reasonable, and (2) designing human enhancers is a relatively challenging task, whereas for yeast promoters, all methods tend to exceed the maximum fitness, we performed a detailed ablation study on our two key contributions—policy warm-started from a pretrained model and the TFBS reward—within the offline MBO setting across all human enhancer tasks. As shown in Table 3:
>
> (1) Pretraining on real sequences proves to be highly beneficial. While the "w/o Pretraining" setup occasionally discovers sequences with high fitness, it underperforms on the Medium metric by 0.03, 0.12, and 0.03 compared to the second-best result across datasets. This demonstrates that pretraining allows the policy to begin in a relatively reasonable exploration space, enabling it to identify a large number of suitable sequences more efficiently. This is particularly advantageous in scenarios like CRE optimization, where large-scale experimental validation can be conducted simultaneously.
>
> (2)The impact of the TFBS Reward can be thoroughly observed and analyzed. Specifically, we have included an ablation study on the effect of the TFBS Reward shown in Table 3. The "w/o TFBS Reward" setup corresponds to α=0, while our TACO model uses a default α=0.01. Additionally, we have also provided results for α=0.1 for further comparison Incorporating the TFBS reward significantly enhances the Medium performance of TACO, achieving best results across all datasets. The method consistently outperforms the w/o TFBS Reward	 baseline by margins of 0.02, 0.01, and 0.02, respectively. These prior-informed rewards guide the policy to explore a more rational sequence space efficiently. Moreover, the biologically guided TFBS Reward is surrogate-agnostic, with the potential to achieve a similar effect to the regularization applied to surrogates in [1], by avoiding excessive optimization towards regions where the surrogate model gives unusually high predictions. The differences in the top fitness and diversity achieved by various models are relatively minor, with no consistent conclusion. Additionally, as the αincreases from the default value of 0.01 to 0.1, our method shows improved performance in both Top and Medium metrics for K562 and SK-N-SH datasets. However, this improvement comes at the cost of a rapid drop in diversity. Interestingly, all metrics for the HepG2 dataset worsen as α grows. We hypothesize that this discrepancy arises from the TFBS Reward, precomputed using the LightGBM model, varying across datasets. Therefore, we recommend carefully tuning α in real-world scenarios to balance the trade-offs effectively.
>
> The updated ablation reuslts are in Section 4.4 Table 5 in the revision.providing a clearer understanding of the method's key contributions.
>
> **Table 3** Ablation study.
> | Dataset   | Setting              | Top ↑        | Medium ↑     | Diversity ↑     | Emb Similarity ↓ |
> |-----------|----------------------|--------------|--------------|-----------------|------------------|
> | **HepG2** | TACO (α = 0.01)      | **0.69 (0.03)** | **0.60 (0.05)** | **141.2 (1.92)** | 0.82 (0.05)       |
> |           | w/o Pretraining      | 0.68 (0.00)  | 0.55 (0.02)  | 139.4 (2.30)    | 0.69 (0.02)       |
> |           | w/o TFBS Reward      | 0.66 (0.05)  | 0.58 (0.07)  | 140.8 (1.64)    | **0.81 (0.05)**   |
> |           | α = 0.1              | 0.65 (0.06)  | 0.56 (0.08)  | 138.6 (3.21)    | 0.86 (0.04)       |
> | **K562**  | TACO (α = 0.01)      | 0.75 (0.09)  | 0.72 (0.10)  | 102.6 (20.14)   | 0.97 (0.04)       |
> |           | w/o Pretraining      | 0.66 (0.15)  | 0.59 (0.16)  | 103.6 (25.77)   | **0.83 (0.14)**   |
> |           | w/o TFBS Reward      | 0.76 (0.07)  | 0.71 (0.08)  | **106.2 (20.90)**| 0.94 (0.05)       |
> |           | α = 0.1              | **0.78 (0.01)** | **0.77 (0.01)** | 82.8 (4.02)     | **0.99 (0.00)**   |
> | **SK-N-SH** | TACO (α = 0.01)    | 0.68 (0.08)  | 0.62 (0.08) | 121.4 (7.86)    | 0.90 (0.03)       |
> |           | w/o Pretraining      | 0.69 (0.02) | 0.57 (0.06)  | **131.8 (11.17)**| **0.74 (0.11)**   |
> |           | w/o TFBS Reward      | 0.67 (0.06)  | 0.60 (0.06)  | 111.6 (12.86)   | 0.89 (0.04)       |
> |           | α = 0.1              | **0.71** (0.01)  | **0.65** (0.02)  | 121.2 (5.45)    | 0.90 (0.05)       |

---

> ### Author Response · Authors · 2024-11-25
> **Response to Reviewer BjdH (5)**
>
> **Q5**: Figures 3A and 5A are below acceptable quality for a publication at ICLR. Both appear to be screenshots and contain either unreadable or poorly labeled axes/subplots.
>
> **A5**: We apologize for the errors that led to issues with the captions or axis labeling. Figure 3A is not a screenshot; it was included in its draft version without updating the early draft caption properly. As for Figure 5A, it is indeed a screenshot from a WandB experiment. Since we have conducted more comprehensive ablation studies in Section 4.4 in the revision, we have removed Figure 5A and retained only Figure 5B as the new Figure 5. All minor errors in the figures have been corrected in the revision.
>
> **Q6**: The References section contains incorrect and inconsistent citation formatting. In particular, many titles are lower case when they should be upper case and inconsistent information is included in the citation (e.g. sometimes URLs are provided and other times they are not). Also, many citations refer to an arXiv pre-print, rather than the published version of a paper; this must be checked and fixed.
>
> **A7**: Thank you for pointing these out. We have made the necessary changes and updated them in the revision. Specifically, we have removed numerous URLs and ensured that some of the most recently published papers were correctly cited, e.g., [3][4].
>
>
> **Q8**: Since TFBS motif design is a task that is of interest to the community (as is mentioned in Related Work), the form of the reward may limit the practical applicability of the method. Can the author's clarify whether their method is able to design novel TFBS motifs? If not, can the author's clarify whether this is an important limitation or not?
>
> **A8**: Thank you for pointing this out. We will clarify below, from four perspectives, why our current design does not have significant limitations.
>
> (1) We utilize an existing database [5] of TFBS motifs that are only known to bind transcription factors, rather than directly using confirmed activatory or repressive TFBS motifs. Subsequently, we infer the cell-specific roles of TFBS motifs in a data-driven manner. This already constitutes a relatively large motif exploration space, enabling meaningful motif design and discovery.
>
> (2) The proposed TFBS reward does not impose a hard constraint on the generated sequences. Instead, it provides a soft reward in addition to the oracle reward: the more similar the motifs contained in the sequences are to some known motifs and the higher the activities of these known motifs (inferred in a data-driven manner using the LightGBM model), the higher the reward. The TFBS reward softly encourages the generated motif to be similar to prior motifs with high activities, but it does not require the generated motif to be exactly the same as the known motif. Since using only the oracle reward can produce sequences that are OOD of the training set (in this situation, the oracle's predicted activity may not be reliable), the TFBS reward can be viewed as a kind of regularization [1] to control the optimization from deviating too much from the known realistic sequences. We have added detailed discussions on this topic to Appendix K in the revision.
>
> (3) Initially, we intended not to rely on pre-defined motifs from databases. Instead, our goal was to iteratively learn potential motifs in a data-driven manner and use these motifs to enhance the fitness of generated sequences, similar to the idea behind the EM algorithm, which has been explored in molecule optimization [6]. However, while extracting motifs from molecular graphs is relatively straightforward due to their clear structural boundaries, DNA sequences lack explicit boundaries, making it significantly more challenging to automatically identify meaningful motifs. Nevertheless, recent advancements in understanding promoter mechanisms [7] may provide valuable insights for revisiting this idea. That said, even in molecule optimization, where advanced automatic motif mining methods [8][9] are available, the use of pre-defined motifs has been consistently demonstrated to be highly effective [10][11]. Therefore, we do not view the reliance on pre-defined motifs as a significant limitation. We have added the detailed related discussions to Appendix K in the revision.
>
> (4)Furthermore, to the best of our knowledge, our work is the first to integrate such essential prior information (TFBS motifs) into the machine learning-driven CRE generation process. Our results demonstrate the effectiveness of incorporating prior knowledge, paving the way for future studies to explore more advanced approaches, such as designing algorithms for automatic motif mining. We believe this will drive further progress and innovation within the community.

---

> ### Author Response · Authors · 2024-11-25
> **Response to Reviewer BjdH (6)**
>
> **References**
>
> [1] Reddy, Aniketh Janardhan, et al. "Designing Cell-Type-Specific Promoter Sequences Using Conservative Model-Based Optimization." bioRxiv (2024).
>
> [2] Angermueller, Christof, et al. "Model-based reinforcement learning for biological sequence design." *International conference on learning representations.* 2019.
>
> [3] Nguyen, Eric, et al. "Sequence modeling and design from molecular to genome scale with Evo." Science 386.6723 (2024): eado9336.
>
> [4] Gosai, Sager J., et al. "Machine-guided design of cell-type-targeting cis-regulatory elements." Nature (2024): 1-10.
>
> [5] Castro-Mondragon, Jaime A., et al. "JASPAR 2022: the 9th release of the open-access database of transcription factor binding profiles." *Nucleic acids research* 50.D1 (2022): D165-D173.
>
> [6] Chen, Binghong, et al. "Molecule optimization by explainable evolution." *International conference on learning representation* (ICLR). 2021.
>
> [7] Dudnyk, Kseniia, et al. "Sequence basis of transcription initiation in the human genome." *Science* 384.6694 (2024): eadj0116.
>
> [8] Kong, Xiangzhe, et al. "Molecule generation by principal subgraph mining and assembling." *Advances in Neural Information Processing Systems* 35 (2022): 2550-2563.
>
> [9] Geng, Zijie, et al. "De Novo Molecular Generation via Connection-aware Motif Mining." The Eleventh International Conference on Learning Representations.
>
> [10] Zhang, Zaixi, et al. "Motif-based graph self-supervised learning for molecular property prediction." *Advances in Neural Information Processing Systems* 34 (2021): 15870-15882.
>
> [11] Wu, Zhenxing, et al. "Chemistry-intuitive explanation of graph neural networks for molecular property prediction with substructure masking." *Nature Communications* 14.1 (2023): 2585.

---

> > ### Comment · Reviewer_BjdH · 2024-11-26
> >
> > I thank the authors for their clear and thorough response. Many of my concerns have been addressed in the updated draft and I have adjusted my score accordingly. I have two remaining concerns
> >
> > 1. I commend the authors' efforts in addressing my primary concern about evaluation in an offline MBO setting and the inclusion of the new Section 4.3 strengthens the paper. However, the authors test of Gradient Ascent gives me pause about their offline MBO setup. In a typical offline MBO setting, GA will tend to produce adversarial examples and thus perform quite poorly. In this case, it performs well which suggests that the surrogate is a good model for the oracle, even for sequences produced after many steps of GA. This may result if, for instance, D_offline is sampled uniformly at random from D and/or if |D_offline| is not much smaller than |D|. Can the authors please clarify how they constructed D_offline? At the very least, the GA result should be included and discussed in the paper.
> > 2. I would like to see the Related Work section further improved to recognize other contributions. I disagree that Angermueller et al. (2019) "did not focus on DNA-related tasks" simply because they also tested on protein tasks. This generalizability is an positive aspect of their method if anything. I think it would be more appropriate for the authors to explain how they build on this previous work (e.g. by adding domain-specific regularization) rather than claiming it is distinct. Further, while I understand the reasons for the authors being unable to compare to Reddy et al. (2024), but I don't think the Related Work sufficiently recognizes their contribution. It should specify that this work also works on the CRE design problem specifically and should also clarify that the authors use in-vitro experiments as their final evaluation (rather than only oracle models, as suggested in the current draft). Then, the authors could describe the various advantages of their approach (i.e. no need for a differentiable surrogate) and why they were unable to compare directly.

---

> > > ### Author Response · Authors · 2024-11-28
> > > **Response to Reviewer BjdH (7)**
> > >
> > > Thank you for your quick feedback and further suggestions. We have made additional modifications to the draft and uploaded the revised version. The newly updated sections are highlighted in **orange**. Below, we address your specific concerns in detail:
> > >
> > > **Q1**: However, the authors' test of Gradient Ascent gives me pause about their offline MBO setup. In a typical offline MBO setting, GA will tend to produce adversarial examples and thus perform quite poorly. In this case, it performs well, which suggests that the surrogate is a good model for the oracle, even for sequences produced after many steps of GA. This may result if, for instance, \(D_{\text{offline}}\) is sampled uniformly at random from \(D\) and/or if \(|D_{\text{offline}}|\) is not much smaller than \(|D|\). Can the authors please clarify how they constructed \(D_{\text{offline}}\)? At the very least, the GA result should be included and discussed in the paper.
> > >
> > > **A1**:
> > >
> > > *Note: In this rebuttal forum, we refer to Gradient Ascent as GA, but in the paper, we consistently use GAs to distinguish it from Genetic Algorithms.*
> > >
> > >
> > > **Minor Correction**: In our initial implementation of GA, we did not restrict the optimization to the softmax-induced one-hot encoded simplex [1]. Instead, the generated encodings were directly hard-clipped between 0 and 1. Our latest GA experiments have addressed and corrected this issue, and overall, the softmax-induced approach demonstrates better performance.
> > >
> > > 1. First, we introduce the details of **how we constructed the offline MBO setting.**
> > >
> > > After incorporating the offline MBO setup as your suggestion, we primarily referred to the settings used in two recent papers. Specifically:
> > >
> > > - [1] did not impose specific fitness quantiles but rather focused on using a different train-validation-test split for the surrogate and oracle.
> > > - [3] divided the complete dataset into two subsets: one for training the surrogate and the other for training the oracle and further restricted the surrogate's training data to fitness values below the 95th percentile, simulating a real-world offline dataset that may lack observations of extremely high fitness values.
> > >
> > > We adopted the approach in [3]. The detailed procedure was already described in Appendix J in the previous revision, and in this version, we have emphasized it further in the main text. Specifically, The sub-sampling strategy involves:
> > >
> > > > Randomly splitting the dataset in half and selecting sequences with fitness values below the 95th percentile to simulate a realistic scenario where observed data may have an upper limit.
> > >
> > > 2. Next, we attempt to discuss **why does GA not perform poorly?**
> > >
> > > This is indeed a surprising observation. To the best of our knowledge, prior CRE design work has not extensively explored GA methods, except for [1]. However, [1] does not seem to include an ablation study on regularization terms (*if we are mistaken, please correct us*). Therefore, in the context of DNA CRE design—where Enformer-based models [2] are widely used to train scoring functions—it remains an open question whether directly applying Gradient Ascent to a differentiable surrogate would result in adversarial examples with poor performance.
> > >
> > > We acknowledge that in the case of a perfect oracle, adversarial examples would likely emerge. However, due to the simple data partitioning strategies commonly used in this field, it appears that a surrogate trained on a subset can sufficiently approximate the oracle.

---

> > > ### Author Response · Authors · 2024-12-02
> > > **Kind Reminder to Review Our Follow-Up Response**
> > >
> > > Dear Reviewer,
> > >
> > > Thank you for the time, effort, and expertise you have invested in reviewing our paper. We also appreciate your follow-up on our initial rebuttal, your thoughtful new suggestions, and your generosity in raising your score.
> > >
> > > We have further conducted experiments to better understand the effect of Gradient Ascent (GA) on our dataset, revealing discrepancies between oracle and surrogate predictions. Additionally, we have included results for different methods under a more challenging setting (60th percentile) in the appendix of the revised version.
> > >
> > > We also acknowledge the significant contributions of previous work by Reddy et al. (2024) and Angermueller et al. (2019). In Section 2 of the revision, we emphasized that Angermueller et al. (2019) developed a generalizable approach for biological sequence design. We also highlighted the algorithmic contributions of Reddy et al. (2024) and, beyond their algorithm, their in vitro validation of the designed promoters, demonstrating the potential of machine learning techniques in CRE design. Additionally, we have clearly outlined how our approach builds upon these foundational works. In Section 4, we also elaborated on why a direct comparison with Reddy et al. (2024) was not feasible.
> > >
> > > As the deadline for the author-reviewer discussion is now less than a day away, we are reaching out to gently remind you to review our latest response.
> > >
> > > We sincerely welcome any further feedback and comments that can help refine our work. If you find that our updates have addressed your concerns, we would be deeply grateful if you could consider further raising your score.
> > >
> > > Thank you again for your time and consideration.
> > >
> > > Sincerely,
> > >
> > > The Authors

---

> > > ### Author Response · Authors · 2024-12-03
> > > **Clarifying Citation Placement and Following Up on Final Feedback**
> > >
> > > We sincerely thank you for your review of our paper. Your suggestions, especially regarding the addition of offline MBO experiments, have significantly enhanced our work. Here, we would like to provide two additional clarifications.
> > >
> > > ### **Clarifying Citation Placement**
> > > When adding the offline MBO experiments, we adopted the 95th percentile surrogate training setup following [1]. While we have already cited this work in the appendix and in the rebuttal forum, we acknowledge that its citation was unintentionally omitted in the main manuscript. We assure you that the proper citation will be included in Section 4.3 of the final version of the manuscript. The method proposed in [1], similar to [2], requires training an additional conservative reward model. This contrasts with the baselines in our manuscript, none of which require this extra component. Therefore, we did not directly compare with [1] but primarily referred to their offline MBO data partitioning strategy.
> > >
> > > ### **Following Up on Final Feedback**
> > > In addition to addressing this oversight, we would like to politely remind you that the review discussion deadline is approaching. We would appreciate knowing whether our responses have resolved your concerns and if you have any further feedback for us to address. If there are any issues, we still have time to provide additional responses and clarifications.
> > >
> > > References
> > >
> > > [1] Uehara, Masatoshi, et al. "Bridging Model-Based Optimization and Generative Modeling via Conservative Fine-Tuning of Diffusion Models." arXiv preprint arXiv:2405.19673 (2024).
> > >
> > > [2] Reddy, Aniketh Janardhan, et al. "Designing Cell-Type-Specific Promoter Sequences Using Conservative Model-Based Optimization." bioRxiv (2024).

---

> ### Author Response · Authors · 2024-11-28
> **Response to Reviewer BjdH (8)**
>
> To further address your concern, we validated GA performance across different fitness (95, 80, 60) quantiles using K562 cells (our default setting was the 95th percentile). *Note that the percentile determines only the training data used for the surrogate, while all methods share the same oracle for evaluation.* We reported the scores predicted by both the surrogate and oracle for the one-hot-encoded simplex (referred to as **Prob**) and the hard-decoded sequences optimized (referred to as **Sequence**) in each iteration. The results for the three quantiles are provided in the revision's Appendix N, Figure 9-11. Our findings indicate:
>
>
> - For the 95th percentile, as illustrated in Appendix Figure 9 in the revision, the fitness in the sequence space initially rises sharply but subsequently drops. Similarly, for the 60th percentile, as depicted in Figure 11, a comparable pattern emerges in the oracle's predictions within the Prob space. These observations highlight **a gap between the surrogate and the oracle**, as the surrogate's predictions consistently increase throughout. This outcome aligns with our expectations in the offline MBO setting, where the surrogate cannot perfectly approximate the oracle.
>
> - However, the oracle's predictions do show significant improvement at the start, indicating that directly applying GA to the surrogate can still benefit the oracle's results. This suggests that, under the current CRE data partitioning strategy, even a surrogate trained on low-fitness subsets can reasonably capture the trends of the oracle’s predictions (although the surrogate itself, having never encountered high-fitness data, predicts much lower upper bounds). This is an interesting question for future research in CRE design. However, we emphasize that our primary focus is on designing optimization algorithms rather than relying on **a differentiable surrogate**. Our current offline MBO setting has already made the task more challenging, achieving the intended goal of designing an offline MBO setting. Nevertheless, we do not yet fully understand why GA does not lead to significantly poor results. We have added these results to Section 6 and Appendix N in the revision for further clarification.
>
> Besides, we have also added the results of different methods (including GA) guided by a surrogate trained on the 60th percentile in Appendix N (Tables 17-19)  in the revision. (We chose the 60th percentile because its fitness threshold is already very low, providing a challenging scenario for evaluation.) It can be observed that, despite the significant gap between the surrogate and the oracle under the 60th percentile training, GA still achieves relatively good performance. Notably, under the 60th percentile setting, PEX, which performed well at the 95th percentile, shows moderate results, while CMAES, which previously performed the worst, achieves excellent performance. Our TACO, in this setting, continues to maintain SOTA results.
>
> Table 1. K562 Normalized Fitness Quantiles
> | Quantile   | Normalized Fitness Value |
> |------------|---------------------------|
> | 60%        | 0.37                      |
> | 80%        | 0.41                      |
> | 90%        | 0.47                      |
> | 95%        | 0.53                      |
>
>
> **Q2**: I would like to see the Related Work section further improved to recognize other contributions.
>
> **A2**:  Thank you for your suggestion. We highly appreciate the significant contributions made by [1] and [4] and acknowledge that we missed highlighting some of their contributions in the previous version. In response, we have further emphasized their specific contributions in Section 2, as well as highlighted the additional contributions our work builds upon their foundations. We have used **orange** text to emphasize the contributions of [1] and [4] in the Section 2 of the revision . Additionally, we have discussed [1] in more detail in Section 4.1.
>
>
> **References**
>
> [1] Reddy, Aniketh Janardhan, et al. "Designing Cell-Type-Specific Promoter Sequences Using Conservative Model-Based Optimization." *bioRxiv* (2024).
>
> [2] Avsec, Žiga, et al. "Effective gene expression prediction from sequence by integrating long-range interactions." *Nature methods* 18.10 (2021): 1196-1203.
>
> [3] Uehara, Masatoshi, et al. "Bridging Model-Based Optimization and Generative Modeling via Conservative Fine-Tuning of Diffusion Models." *arXiv preprint* arXiv:2405.19673 (2024).
>
> [4] Angermueller, Christof, et al. "Model-based reinforcement learning for biological sequence design." *International conference on learning representations.* 2019.

---

> ### Author Response · Authors · 2024-11-30
> **Follow-Up on Second-Round Discussion and Feedback**
>
> Dear Reviewer,
>
> Thank you for your quick feedback on our first-stage response and for kindly raising your score. We also greatly appreciate your remaining concerns, which help us further improve the quality of our paper.
>
> As the discussion deadline is approaching, we kindly request further feedback from you to help us refine the quality of our paper and address any remaining concerns. Your insights are highly valued and will play a crucial role in enhancing the clarity and impact of this work.
>
> We have made the following key revisions:
> - Conducted more in-depth discussions and experiments exploring the performance of GA on our dataset, while also including comparisons under more challenging MBO settings.
> - Emphasized the contributions of prior works in the revision and clarified how our work builds upon their foundations.
>
> We understand that it is currently the Thanksgiving period (if applicable to you), and we apologize for any inconvenience caused by this message. We wish you a Happy Thanksgiving and sincerely thank you for your efforts in fostering progress within the research community.
>
> Best regards,
> The Authors

---

### Official Review · Reviewer_tdUH · 2024-11-03

**Soundness:** 3
**Presentation:** 3
**Contribution:** 3
**Rating:** 8
**Confidence:** 4

**Summary:**

The paper presents a novel method for designing cis-regulatory elements (CREs) using a reinforcement learning (RL) framework that enables the generation of high-fitness, cell-type-specific, and diverse sequences. The primary objective is to produce CRE sequences that are capable of enhancing gene expression. Current methods, often reliant on greedy algorithms or directed evolution approaches, lack biological insight for guiding the exploration of sequence space. These traditional methods tend to get trapped in local minima, resulting in CREs that are limited in diversity and difficult to interpret biologically.

Main Contributions

Development of an RL Framework: The authors introduce a reinforcement learning framework that fine-tunes a pre-trained autoregressive generative model, HyenaDNA, for designing high-fitness, cell-type-specific CREs. This approach also aims to maintain sequence diversity, addressing limitations in traditional methods.

Integration of TFBS Modulation: The RL process actively incorporates transcription factor binding sites (TFBS) to encourage biologically meaningful changes, such as removing repressor motifs and adding activator motifs. This guidance helps enhance the regulatory potential of the generated sequences.

Evaluation Across Multiple Contexts: The proposed method is tested across different tasks:
An enhancer design task for three distinct human cell types.
A promoter design task across two yeast species in varied media conditions, demonstrating the framework’s adaptability to different biological contexts.

**Strengths:**

This paper makes several contributions to the field of DNA sequence design:

Biologically Guided Sequence Design: By integrating TFBS modulation into the RL framework, the approach maintains biological relevance, encouraging the addition of activators and removal of repressors, which enhances the model's potential for generating functional CREs.

Innovation in CRE Design: The reinforcement learning paradigm offers a new way to explore the sequence space more effectively, overcoming the limitations of greedy and evolution-based methods that often produce low-diversity sequences.

Comprehensive Evaluation: The authors rigorously evaluate the framework’s performance on both enhancer and promoter design tasks, providing evidence for its flexibility and applicability to various regulatory elements and cell types.

Overall, the paper is written well and of good quality.

**Weaknesses:**

The paper would benefit from a more extensive Discussion section to contextualize the results further.

Minor Points
Include references to Appendix B/C in the introduction or caption of Table 1.
In the preliminary experiment described in the introduction, add a reference to Appendix E or specify the number of TFBS scanned for each model.
In Figure 2, clarify the meaning of the BOS token and C or T action symbols.
Figure 3 is missing "A" and "B" labels in the visuals.
In Appendix F, correct the sentence, "Our results indicate that only the metric has a significant impact on the final performance. The ablation results are summarized in Table 7."
Explicitly describe the regularization technique (entropy regularization) discussed in Section 4.3 and reference it in the RL Implementation Details section.
Section 4.3 might be better suited as an appendix to improve readability, and increase space for a larger discussion/conclusion.

**Questions:**

1.What motivated the choice of the HyenaDNA model beyond it being the only published autoregressive DNA language model? Is its receptive field size of 1 million excessive for training on CREs, given that the DNA sequence lengths are only 80 and 200?

2. How was the yeast Enformer model trained? Specifically, what sequences were used, and what was the target variable for regression?
Why do the fitness percentiles selected for D range from 20-80%? Why not use the full range, such as 10-90%, to potentially capture a wider diversity?

3. Did you consider using cosine similarity as a distance metric when training the LightGBM models?

4. Have you thought about adding a feature in the LightGBM model to consider interactions between two TFBS, requiring that they be present together or at a specific distance? This could capture complex TFBS interactions as discussed in Georgakopoulos-Soares et al., "Transcription factor binding site orientation and order are major drivers of gene regulatory activity" (Nature Communications, 2023). (This could be added to the discussion.)

5.Could you clarify whether a specific threshold or adaptive strategy is used to define "low likelihood" sequences in the regularization approach, and how this affects the exploration-exploitation balance in sequence generation?

6.Why did you choose six sequences for the "Top" evaluation metric, and why generate a total of 128 sequences?

7. Could it be more informative to report fitness values close to, but not capped at, 1 to show variability among high-fitness sequences? Additionally, could you provide the standard deviation of fitness scores across generated sequences for each method in Table 2? If the standard deviation is 1, could u sample more sequences? Would you also consider reporting on 'Low' or 'Bottom' fitness sequences to better understand model limitations and areas for improvement?

8. Given that diversity appears beneficial for finding novel targets, would continuing beyond 100 iterations lead to further exploration and potentially better results? Additionally, could you elaborate on how initial conditions influence the observed diversity and fitness metrics, and whether different starting points could impact the algorithm’s optimization effectiveness?

---

> ### Author Response · Authors · 2024-11-25
> **Response to Reviewer tdUH (1)**
>
> Thanks for your detailed and insightful comments. We will address each of your concerns point by point in the following response.
>
> **Q1**: The paper would benefit from a more extensive Discussion section to contextualize the results further.
>
> **A1**: Thank you for this suggestion. We have expanded the Discussion section in the revision to better contextualize the results. Besides, We have added more discussions throughout the paper to better contextualize our method, explain why it works, and identify potential limitations.
>
> (1) We have conducted detailed ablation studies on our model's two core contributions: fine-tuning a pretrained DNA autoregressive model with RL and applying the TFBS Reward. These experiments, through thorough ablations, demonstrate why our model works, and the results can be found in Section 4.4 of the revision.
>
> (2) We have included baselines with conditional generative models in Section 4.3 and provided a detailed discussion of the relationship between conditional generative models and optimization-based approaches. This highlights why RL is necessary to fine-tune autoregressive models. Further details are discussed in Appendix L.
>
> (3)Following Reviewer BjdH's suggestion, we incorporated an offline model-based optimization (MBO) setting. In this setting, we assume that only a subset of the offline dataset, containing sequences with low activities, is available for training. The optimization process relies on a surrogate model trained on this subset instead of the oracle trained on the complete dataset. A simple example in Appendix J (Figure 8) illustrates the necessity of this setting. We also evaluated our method and baselines under this setting and observed that all models' performance declined. Therefore, we recommend future benchmarking of optimization algorithms in such realistic and challenging scenarios.
>
> (4) In Section 5, we discuss potential areas for improvement in our current model design, including data-driven methods to automatically identify functional motifs and explicitly modeling interactions between different TFs.
> We hope these additional discussions provide a clearer understanding of our method and pave the way for advancements in the field of DNA design.
>
> **Q2**: Minor points. (1) Include references to Appendix B/C in the introduction or caption of Table 1. (2) In the preliminary experiment described in the introduction, add a reference to Appendix E or specify the number of TFBS scanned for each model. (3) In Figure 2, clarify the meaning of the BOS token and C or T action symbols. (4) In Figure 3, add missing "A" and "B" labels to the visuals. (5) In Appendix F, correct the sentence: "Our results indicate that only the metric has a significant impact on the final performance. The ablation results are summarized in Table 7." (6) Explicitly describe the regularization technique (entropy regularization) discussed in Section 4.3 and reference it in the RL Implementation Details section. (7) Consider moving the original Section 4.3 to an appendix to improve readability and increase space for a larger discussion/conclusion.
>
> **A2**: Thank you for pointing these out. In the revised draft, we have highlighted all the new changes in **blue** for clarity.
>
> (1) References to Appendix B and C have been added in the introduction.
>
> (2) Appendix E has also been referenced in the introduction. We have also added detailed information about the scanned TFBS motifs in Appendix E.1.
>
>  (3) BOS, part of the paradigm used by autoregressive language models for sequence generation, stands for "beginning of sequence" and is a reserved token indicating the start of a sequence. C and T represent nucleotide bases (A, T, C, G). We apologize for the minor issues in the initial version of the figures, e.g., the incorrect display of bases above the action in the figure. These have been corrected, and more detailed captions have been added. This clarification has been included in Figure 2 in the revision.
>
> (4) The caption for Figure 3 has been corrected accordingly.
>
> (5) The sentence has been revised to clarify that only the metric significantly impacts performance, while the learning rate and number of leaves have nearly no effect.
>
> (6) The RL Implementation Details section (renamed to *Supporting RL Designs*) now explicitly describes the entropy regularization technique, with references to the corresponding ablation experiments.
>
> (7) Following your suggestion, we have moved Section 4.3 to Appendix I.2 in the revision.

---

> ### Author Response · Authors · 2024-11-25
> **Response to Reviewer tdUH (2)**
>
> **Q3**: What motivated the choice of the HyenaDNA model beyond it being the only published autoregressive DNA language model? Is its receptive field size of 1 million excessive for training on CREs, given that the DNA sequence lengths are only 80 and 200?
>
> **A3**: Thank you for pointing this out. The primary reason for choosing the HyenaDNA model is indeed that it is the only available and powerful autoregressive DNA language model. The 1M receptive field size of the pretrained HyenaDNA does introduce a gap in performance when applied to our short CRE design tasks. To address this, we started with the pretrained weights of HyenaDNA and fine-tuned the model on relatively short CRE sequences, aligning it better with the task requirements. As shown in the table below, fine-tuning on offline CRE data slightly improves performance. We have provided a more detailed clarification in Section 3.2 and Appendix D.1.
>
> **Table 1: Performance (HepG2 hard) comparison of pretrained and fine-tuned HyenaDNA on short CRE sequences.**
>
> | Model                | Top ↑   | Medium ↑   |
> |----------------------|---------|------------|
> | Pretrained HyenaDNA  | 0.749   | 0.723      |
> | Fine-tuned HyenaDNA  | **0.751** | **0.729**  |
>
> **Q4**: How was the yeast Enformer model trained? Specifically, what sequences were used, and what was the target variable for regression? Why do the fitness percentiles selected for D range from 20-80%? Why not use the full range, such as 10-90%, to potentially capture a wider diversity?
>
> **A4**: Each dataset originates from cell-specific MPRA experiments, where each fixed-length CRE candidate sequence (80 bp for yeast and 200 bp for human) is associated with an experimental fitness measurement. We trained a model based on the Enformer backbone (a hybrid of CNN and Transformer), using these short sequences as input to predict a single scalar fitness value instead of thousands of genomic profiles. For data partitioning, we followed regLM's approach [1], dividing the data into five equal parts and excluding the highest and lowest portions, retaining the 20-80% range to create a more balanced optimization problem. While a wider range (e.g., 10-90%) could capture greater diversity, we initially adhered to RegLM's setting. In our latest draft, we incorporated an updated offline MBO setting (Section 4.3 and Appendix J) to ensure broader coverage and enhanced diversity.
>
>
> **Q5**: Did you consider using cosine similarity as a distance metric when training the LightGBM models?
>
> **A5**: Thank you for your suggestion. LightGBM is used as a regression model in this context, meaning its output is a scalar representing fitness. We are unsure how similarity metrics, such as cosine similarity, would be applied in this scenario. We also noticed that our Appendix F mistakenly mentioned a "distance metric," and we acknowledge this as an error. We have fixed it in the revision.
>
> **Q6**: Have you thought about adding a feature in the LightGBM model to consider interactions between two TFBS, requiring that they be present together or at a specific distance?
>
> **A6**: Thank you for your suggestion. We agree that this is an excellent idea. Initially, we also considered explicitly modeling interactions between TFBS, such as enforcing their co-occurrence or specific spacing constraints. However, we decided not to implement these more complex designs, as autoregressive models can implicitly capture such interactions through their ability to model joint distributions (as in language modeling). Our work represents an initial attempt to integrate prior knowledge of TFBS into CRE design, and we plan to explore more sophisticated approaches to modeling TFBS interactions in future work. Furthermore, we have included a discussion on this topic in the revised manuscript, referencing the work of Georgakopoulos-Soares et al. [2].

---

> ### Author Response · Authors · 2024-11-25
> **Response to Reviewer tdUH (3)**
>
> **Q7**: Could you clarify whether a specific threshold or adaptive strategy is used to define "low likelihood" sequences in the regularization approach, and how this affects the exploration-exploitation balance in sequence generation?
>
> **A7**: Thank you for your question. In our autoregressive model, the policy `π` determines the probability `π(a_i | s)` at each step, where `a_i` represents the next base to be generated. If the RL agent always selects the action with the highest probability from `π`, it risks converging to sub-optimal solutions due to insufficient exploration. To address this, we incorporate entropy regularization to encourage the model to explore actions with lower probabilities. Specifically, this means the agent is motivated to consider less likely sequences by assigning an additional entropy term to the gradient strategy. The entropy regularization term, shown below, is adaptive:  `-1 / log(π(a | s))`. This term dynamically adjusts its influence: when the policy assigns very high probabilities to specific actions, the regularization term grows larger, effectively penalizing overconfidence in these actions. Conversely, it promotes exploration of actions with lower probabilities. This adaptive mechanism helps balance exploration and exploitation, ensuring the generation of diverse sequences while avoiding local optima (Section 3.2 and Appendix I.2).
>
> **Q8**: Why did you choose six sequences for the "Top" evaluation metric, and why generate a total of 128 sequences?
>
> **A8**: First, we acknowledge that "six" is a typo—the correct number is 16, as also reflected in our supplementary material. The choice of 16 sequences for the "Top" evaluation metric and a total of 128 generated sequences was inspired by a study [3] published at ICML 2024, which focused on protein sequence optimization. We have corrected this issue in the revision and included the citation.
>
>
> **Q9**: Could it be more informative to report fitness values close to, but not capped at, 1 to show variability among high-fitness sequences? Additionally, could you provide the standard deviation of fitness scores across generated sequences for each method in Table 2? If the standard deviation is 1, could u sample more sequences? Would you also consider reporting on 'Low' or 'Bottom' fitness sequences to better understand model limitations and areas for improvement?
>
> **A9**: Thank you for your suggestion. In our original setting, we assumed the presence of a perfect oracle, which resulted in overly optimistic and potentially unreliable high-fitness values, particularly for simpler tasks like yeast promoter optimization. To address this limitation, we have supplemented our experiments with results from an offline MBO setting, as shown in Section 4.3 in the revision. This adjustment provides a more realistic evaluation by focusing on sequences within the surrogate's training distribution, which helps mitigate the issue of inflated fitness scores.
>
> While promoter data in this updated setting still occasionally exceeds the maximum, the excess is much smaller. Consequently, we have reported the results for promoter sequences without capping at 1 in Appendix Table 12 and Table 13. Your suggestions, including (1) reporting diversity among sequences with fitness values close to 1, (2) sampling more sequences, and (3) reporting low-fitness sequences to better understand model limitations, are very interesting. We plan to explore these directions in future work to further enhance the robustness and interpretability of CRE sequence design evaluations.

---

> ### Author Response · Authors · 2024-11-25
> **Response to Reviewer tdUH (4)**
>
> **Q10**: Given that diversity appears beneficial for finding novel targets, would continuing beyond 100 iterations lead to further exploration and potentially better results? Additionally, could you elaborate on how initial conditions influence the observed diversity and fitness metrics, and whether different starting points could impact the algorithm’s optimization effectiveness?
>
> **A10**: Thank you for raising this point.
>
> (1) As mentioned in **A9**, when transitioning to the offline MBO setting, we observed that sequences optimized in later iterations often achieve artificially high fitness according to the surrogate model. However, the oracle's actual predictions for these sequences plateau, indicating a divergence between the surrogate and the true oracle. **Figure 8 in Appendix J** provides an intuitive example: as the iterations progress, the surrogate's predicted fitness values continue to increase, while the oracle's predictions hit a bottleneck. This behavior highlights the challenge of maintaining reliable optimization over extended iterations and underscores the importance of controlling for out-of-distribution predictions in surrogate-based methods. This observation also inspires future work on DNA design to focus more on evaluating the plausibility of generated sequences, ensuring that out-of-distribution samples do not mislead the surrogate and compromise optimization effectiveness.
>
> (2) Here, we understand "initial conditions" to refer to two aspects.
>
> - The first is the data partitioning described in Section 4.1. This partitioning has minimal impact on our autoregressive generative model because these initial sequences primarily serve as a good starting replay buffer. The diversity of sequences generated by the autoregressive model during each iteration is already substantial, so the initial sequences do not significantly influence subsequent optimization. However, this is not the case for mutation-based optimization methods, which are heavily dependent on the initial sequences as their starting point for optimization.
>
> - The second aspect concerns the initialization of the policy weights, which is undoubtedly critical. One of our key contributions, pretraining, specifically addresses this issue. A well-initialized policy allows exploration to begin directly in a reasonable (and potentially high-fitness) region of the data space, providing a solid starting point that enables the discovery of high-fitness sequences. As shown in Table 3, pretraining on real sequences proves to be highly beneficial. While the "w/o Pretraining" setup occasionally discovers sequences with high fitness, it underperforms on the Medium metric by 0.03, 0.12, and 0.03 compared to the second-best result across datasets. This demonstrates that pretraining enables the policy to start exploration in a relatively reasonable space, allowing it to identify a large number of suitable sequences more efficiently. This advantage is particularly significant in scenarios like CRE optimization, where large-scale experimental validation can be conducted simultaneously.
>
> **Table 3** Ablation study.
> | Dataset   | Setting              | Top ↑        | Medium ↑     | Diversity ↑     | Emb Similarity ↓ |
> |-----------|----------------------|--------------|--------------|-----------------|------------------|
> | **HepG2** | TACO (α = 0.01)      | **0.69 (0.03)** | **0.60 (0.05)** | **141.2 (1.92)** | 0.82 (0.05)       |
> |           | w/o Pretraining      | 0.68 (0.00)  | 0.55 (0.02)  | 139.4 (2.30)    | 0.69 (0.02)       |
> |           | w/o TFBS Reward      | 0.66 (0.05)  | 0.58 (0.07)  | 140.8 (1.64)    | **0.81 (0.05)**   |
> |           | α = 0.1              | 0.65 (0.06)  | 0.56 (0.08)  | 138.6 (3.21)    | 0.86 (0.04)       |
> | **K562**  | TACO (α = 0.01)      | 0.75 (0.09)  | 0.72 (0.10)  | 102.6 (20.14)   | 0.97 (0.04)       |
> |           | w/o Pretraining      | 0.66 (0.15)  | 0.59 (0.16)  | 103.6 (25.77)   | **0.83 (0.14)**   |
> |           | w/o TFBS Reward      | 0.76 (0.07)  | 0.71 (0.08)  | **106.2 (20.90)**| 0.94 (0.05)       |
> |           | α = 0.1              | **0.78 (0.01)** | **0.77 (0.01)** | 82.8 (4.02)     | **0.99 (0.00)**   |
> | **SK-N-SH** | TACO (α = 0.01)    | 0.68 (0.08)  | 0.62 (0.08) | 121.4 (7.86)    | 0.90 (0.03)       |
> |           | w/o Pretraining      | 0.69 (0.02) | 0.57 (0.06)  | **131.8 (11.17)**| **0.74 (0.11)**   |
> |           | w/o TFBS Reward      | 0.67 (0.06)  | 0.60 (0.06)  | 111.6 (12.86)   | 0.89 (0.04)       |
> |           | α = 0.1              | **0.71** (0.01)  | **0.65** (0.02)  | 121.2 (5.45)    | 0.90 (0.05)       |

---

> ### Author Response · Authors · 2024-11-25
> **Response to Reviewer tdUH (5)**
>
> **References**
>
> [1] Lal, Avantika, et al. "regLM: Designing realistic regulatory DNA with autoregressive language models." *International Conference on Research in Computational Molecular Biology.* Cham: Springer Nature Switzerland, 2024.
>
> [2] Georgakopoulos-Soares, Ilias, et al. "Transcription factor binding site orientation and order are major drivers of gene regulatory activity." *Nature communications* 14.1 (2023): 2333.
>
> [3] Lee, Minji, et al. Robust Optimization in Protein Fitness Landscapes Using Reinforcement Learning in Latent Space. *Forty-first International Conference on Machine Learning.*

---

> ### Author Response · Authors · 2024-11-30
> **Kind Request for Further Feedback Before Discussion Deadline**
>
> Dear Reviewer,
>
> Thank you for recognizing the contributions of our work and for your suggestions on how to further improve it. We greatly appreciate the time and effort you have dedicated to reviewing our paper.
>
> Regarding the weaknesses and questions you pointed out, we have provided detailed discussions and explanations (both in the discussion forum and in the revision) and supplemented additional experiments to strengthen the validation of our method. We also deeply appreciate your suggestions for areas of improvement. These suggestions have been thoroughly discussed and incorporated into the discussion section of our revision, as we believe they will significantly enhance the quality of our work.
>
> As the discussion deadline is approaching, we kindly request further feedback from you to help us refine the quality of our paper and address any remaining concerns. Your insights are highly valued and will play a crucial role in improving the clarity and impact of this work.
>
> We understand that it is currently the Thanksgiving period (if applicable to you), and we apologize for any inconvenience caused by this message. We wish you a Happy Thanksgiving and sincerely thank you for your efforts in fostering progress within the research community.
>
> Best regards,
>
> The Authors

---

### Official Review · Reviewer_2GBK · 2024-11-04

**Soundness:** 3
**Presentation:** 3
**Contribution:** 2
**Rating:** 8
**Confidence:** 5

**Summary:**

In this paper, the authors introduce TACO, a method for designing DNA cis-regulatory elements (CREs). TACO leverages reinforcement learning (RL) fine-tuning of a pretrained DNA autoregressive generative model to maximize CRE activity while maintaining diversity.  The approach involves training an "oracle" model on activity data and extracting transcription factor binding motifs (TF motifs) by interpreting the oracle's extracted features using SHAP values.

TACO formulates the Markov Decision Process (MDP) problem with an empty initial state, where actions involve appending nucleotides to the sequence at each step. Episodes terminate after T steps. The oracle's prediction serves as the final reward, while intermediate non-zero rewards are assigned based on the extracted TF motifs to guide the search process.  Specifically, negative rewards are given for repressive motifs, and positive rewards are given for enhancing motifs.

The authors compare TACO to standard optimization methods using the same oracle on datasets for promoter activity in yeast and enhancer activity in human cell lines.  Their results demonstrate that while TACO does not necessarily yield higher predicted activity for generated CREs, it achieves greater diversity compared to standard optimization methods.

**Strengths:**

- The problem of generating diverse and high-fitness CREs is a crucial area of biological research, and I commend the authors for their contribution to this field.

- I particularly appreciate their exploration of RL fine-tuning, a novel approach in this domain with promising potential, given its success in other areas.

- The authors' meticulous design of the oracle for each benchmark is commendable.

- The automated extraction of TF motifs using SHAP values from the trained LightGBM oracle, coupled with its use in reward shaping for guided search, is an interesting approach.

- The paper is well-written and structured, ensuring clarity and ease of understanding for the reader. The technical details are well-presented, and the comprehensive literature review effectively covers important and recent works.

**Weaknesses:**

The primary weaknesses of this paper lie in the experimental setup and the choice of baselines.

- Across all datasets, the authors report that all methods generate sequences with strong activity. This suggests that the tasks may be "too easy," potentially undermining their suitability for benchmarking generative models like TACO, which aim to generate high-activity sequences.

- While the included baselines are relevant, they rely on older optimization techniques that have been consistently outperformed by modern generative methods, such as autoregressive models and diffusion models, particularly in terms of generating diverse data. Consequently, the reported results, while valid, are not particularly insightful. A more compelling analysis would compare different generative techniques for CRE design. The authors acknowledge in their literature review the emergence of diffusion models (D3 and DNADiffusion) and autoregressive models (regLM) specifically for CRE design. Benchmarking TACO against these methods is essential.

- The introduced diversity metric primarily ensures that generated sequences avoid collapsing to a single mode. While this highlights the known limitations of standard optimization techniques like genetic algorithms, it does not comprehensively evaluate the overall quality of sequences generated by a generative model like TACO. Given that predicted activity appears to be non-discriminative, incorporating additional quality and diversity metrics from a data distribution perspective would be beneficial. I recommend exploring the validation pipeline introduced in D3, which is highly relevant in this context.

- The introduction of reward shaping based on automatically extracted TF motifs is presented as a key contribution; however, the results in Figure 5 do not strongly support its efficacy. Specifically, TACO achieves comparable diversity with alpha=0 (no reward shaping) while maintaining high sequence activity. Including TACO with alpha=0 in Tables 2 and 3 might demonstrate its performance, as it would likely achieve the highest activity while maintaining high diversity, the primary differentiating factor in these studies.

Although the investigated domain and technique are interesting and relevant, and the paper is well-written, the authors do not fully demonstrate the added value of their method. The benchmark also lacks critical baselines. I would be willing to reconsider my assessment if (1) the authors compare their method against at least one of D3, DNADiffusion, and regLM; (2) they introduce additional metrics to evaluate generated sequence diversity; and (3) they provide more convincing evidence for the effectiveness of their reward shaping technique.

**Questions:**

Questions/Comments:

- Algorithm 1 appears too early in the paper, as it references elements and equations introduced later. Moving it further down would improve the flow.

- Equations 1, 3, and 4 could be moved to the appendix, as they are well-known in the field.

- Could the authors provide more details on how the maximum number of steps T is determined?

- Could the authors clarify the following statement: "We set the maximum number of optimization iterations to 100, with up to 256 oracle calls allowed per iteration." Is one optimization iteration equivalent to one episode? If so, does "256 oracle calls" imply that T=256?
Why did the authors retrain HyenaDNA from scratch on D_star? Why not start from a pretrained HyenaDNA model?

- The authors claim a lack of strong encoder backbones like ESM in the DNA field. This seems inaccurate, given recent publications like Nucleotide Transformer, DNABert, HyenaDNA, Caduceus, Evo, Borzoi, Enformer, and many others. If the authors disagree with the claims from these papers, they should provide a detailed argument.

- Using "medium" for both a dataset and a metric could be confusing.

- The authors frequently cite Almeida et al. but do not use their fly enhancer data in the evaluation. What motivated this decision?

- At the end of the "conditional DNA generative models" section, the authors state: "However, these generative methods are designed to fit existing data distributions, limiting their ability to design sequences that have yet to be explored by humans." However, TACO seems to suffer from the same limitation, as its exploration space is bounded by the trained oracle, which is itself limited by the available data distribution. I would appreciate the authors' perspective on this.

---

> ### Author Response · Authors · 2024-11-25
> **Response to Reviewer 2GBK (1)**
>
> Thanks for your detailed and insightful comments. We will address each of your concerns point by point in the following response.
>
> **Q1**: This suggests that the tasks may be "too easy," potentially undermining their suitability for benchmarking generative models like TACO, which aim to generate high-activity sequences.
>
> **A1**: Thank you for pointing out this. The "easy" tasks may be due to the experimental setting, particularly regarding the yeast dataset, where the oracle trained on the complete dataset is used to guide the generation process and leads to overestimated activities. In our revision, following Reviewer BjdH's suggestion, we have incorporated an offline model-based optimization (MBO) setting. In this setting, we assume that only a subset of each offline dataset is available for training, where sequences in this subset have low activities. The optimization process is guided by a surrogate model trained on the subset, rather than relying on the oracle trained on the complete dataset. Since the surrogate model is trained using only sequences with low activities, the activities of some generated sequences predicted by the oracle model beyond the training activity range may not be accurate. Therefore, the final evaluation is performed on the remaining set and the evaluation results are measured using the oracle model trained on the full dataset. Since sequences with higher activities are also included in the training set, the oracle model can more accurately evaluate the activities of the generated sequences. In the setting, the performance of all methods significantly decreases. However, the relative relationships among the models remain consistent with previous observations. Our TACO achieves the highest Diversity while maintaining fitness optimization comparable to the best-performing methods.  Results on other datasets can be found in Appendix M of the revised draft. Appendix J Figure 8 provides an example where the surrogate is misled.
>
> **Table 1** Offline MBO results for human enhancers (K562).
> | **Model**   | **Top ↑**  | **Medium ↑**  | **Diversity ↑**  | **Emb Similarity ↓** |
> |-------------|------------|---------------|------------------|----------------------|
> | **PEX**     | **0.76 (0.02)** | **0.73 (0.02)** | 15.8 (4.97)     | 0.97 (0.01)         |
> | **AdaLead** | 0.66 (0.08) | 0.58 (0.06)    | 63.2 (70.01)     | 0.88 (0.12)         |
> | **BO**      | 0.71 (0.07) | 0.64 (0.08)    | 43.6 (6.91)      | 0.87 (0.04)         |
> | **CMAES**   | 0.66 (0.02) | 0.44 (0.03)    | 79.2 (3.83)      | **0.35 (0.03)**     |
> | **regLM**   | 0.69 (0.02) | 0.47 (0.01)    | **149.60 (0.49)**| 0.38 (0.02)     |
> | **DDSM**    | 0.43 (0.00) | 0.40 (0.00)    | 93.40 (0.49)     | 0.80 (0.00)         |
> | **TACO**    | 0.75 (0.09) | 0.72 (0.10)    | 102.6 (20.14)| 0.97 (0.04)         |
>
> **Q2**:  The authors acknowledge in their literature review the emergence of diffusion models and autoregressive models specifically for CRE design. Benchmarking TACO against these methods is essential.
>
> **A2**: Thank you for this suggestion. We have added comparisons with the latest generative models, including the autoregressive generative model regLM [1] and the discrete diffusion model DDSM [2] (The primary reason for selecting DDSM is that it is the first work on DNA diffusion models, and its well-maintained codebase makes it highly reproducible.). We adopted conditional generation to evaluate the generative models. Specifically, regLM used the official pretrained weights, and sequences were generated based on the prefix label with the highest fitness score in each dataset. DDSM, on the other hand, was trained on our offline data, where labels for data points above the 95th percentile were set to 1 and others to 0. A conditional diffusion model was then trained using these labels, and sequences were generated with 1 as the condition for evaluation. As shown in Table 1, our method outperforms the generative model baselines on fitness-related metrics across all datasets. It is important to note that since these generative models are designed to fit the observed data distribution, their fitness scores are typically lower than the maximum values in the dataset. Additionally, regLM directly used the official pretrained weights, which might have been exposed to data with higher fitness scores than our offline data, but even so, it fails to outperform optimization-based methods. Results on other datasets can be found in Appendix M of the revised draft, and a detailed discussion of generative model performance is provided in Appendix L.

---

> > ### Author Response · Authors · 2024-11-25
> > **Response to Reviewer 2GBK (6)**
> >
> > **References**
> >
> >
> > [1] Lal, Avantika, et al. "regLM: Designing realistic regulatory DNA with autoregressive language models." *International Conference on Research in Computational Molecular Biology.* Cham: Springer Nature Switzerland, 2024.
> >
> > [2] Avdeyev, Pavel, et al. "Dirichlet diffusion score model for biological sequence generation." *International Conference on Machine Learning.*
> >
> > [3] Sarkar, Anirban, et al. "Designing DNA With Tunable Regulatory Activity Using Discrete Diffusion." *bioRxiv*.
> >
> > [4] Reddy, Aniketh Janardhan, et al. "Designing Cell-Type-Specific Promoter Sequences Using Conservative Model-Based Optimization." bioRxiv (2024).
> >
> > [5] Raj Ghugare, et al. "Searching for High-Value Molecules Using Reinforcement Learning and Transformers." *The Twelfth International Conference on Learning Representations*.
> >
> > [6] Yu, Tianhao, et al. "Enzyme function prediction using contrastive learning." *Science* 379.6639 (2023): 1358-1363.
> >
> > [7] Tang, Z., et al. "Evaluating the representational power of pre-trained DNA language models for regulatory genomics." *bioRxiv*.
> >
> > [8] Meier, Joshua, et al. "Language models enable zero-shot prediction of the effects of mutations on protein function." *Advances in neural information processing systems* 34 (2021): 29287-29303.
> >
> > [9] Benegas, G., et al. GPN-MSA: An alignment-based DNA language model for genome-wide variant effect prediction. *bioRxiv*.
> >
> > [10] Huang, C., et al. Personal transcriptome variation is poorly explained by current genomic deep learning models. *Nature Genetics, 55*(11), 1494–1500.
> >
> > [11] Karollus, A., et al. Current sequence-based models capture gene expression determinants in promoters but mostly ignore distal enhancers. *Genome Biology, 24*(56).
> >
> > [12] de Almeida, Bernardo P., et al. "DeepSTARR predicts enhancer activity from DNA sequence and enables the de novo design of synthetic enhancers." *Nature genetics* 54.5 (2022): 613-624.
> >
> > [13] Somepalli, Gowthami, et al. "Understanding and mitigating copying in diffusion models." *Advances in Neural Information Processing Systems* 36 (2023): 47783-47803.

---

> ### Author Response · Authors · 2024-11-25
> **Response to Reviewer 2GBK (2)**
>
> **Q3**: Given that predicted activity appears to be non-discriminative, incorporating additional quality and diversity metrics from a data distribution perspective would be beneficial.
>
> **A3**: Thank you for your suggestion. We value the metrics proposed by D3 [3] for assessing the quality of DNA sequence generation. These metrics are primarily tailored for generative models, focusing on comparing generated data to the real data distribution, where better alignment indicates superior performance. However, since our setting is optimization-based, many of these metrics are not directly applicable. To bridge this gap, we have adapted one of D3's distribution-based metrics: using the oracle to compute sequence embeddings and calculating the mean pairwise cosine similarity of the proposed sequences' embeddings. We refer to this metric as **Emb Similarity**. This metric is particularly suitable for the offline MBO setting, where the oracle does not guide the optimization process, allowing the embeddings produced by the oracle to serve as a fair measure of the data distribution.
>
> As shown in Table 2, TACO's Emb Similarity is lower than other optimization methods, indicating that TACO achieves diversity not only at the sequence level but also at the feature level. However, generative models such as regLM [1] and DDSM [2] exhibit significantly lower Emb Similarity values. For CMAES, this metric is the lowest across most datasets. The primary reason is that the sequences optimized by CMAES tend to have low fitness, which may render them out-of-distribution (OOD) for the oracle, resulting in highly diverse embeddings.
>
> **Table 2** Emb Similarity across different methods and datasets.
> | Dataset  | PEX        | AdaLead    | BO         | CMAES      | regLM      | DDSM      | TACO      |
> |----------|------------|------------|------------|------------|------------|-----------|-----------|
> | Complex  | 0.98(0.01) | 0.95(0.00) | 0.97(0.01) | 0.75(0.05) | 0.91(0.01) | 0.81(0.01) | 0.93(0.01) |
> | Defined  | 0.98(0.01) | 0.98(0.01) | 0.97(0.01) | 0.59(0.05) | 0.90(0.00) | 0.86(0.01) | 0.97(0.01) |
> | HepG2    | 0.98(0.01) | 0.84(0.16) | 0.83(0.13) | 0.45(0.04) | 0.28(0.02) | 0.99(0.00) | 0.82(0.05) |
> | K562     | 0.97(0.01) | 0.88(0.12) | 0.87(0.04) | 0.35(0.03) | 0.38(0.02) | 0.80(0.00) | 0.97(0.04) |
> | SK-N-SH  | 0.98(0.01) | 0.96(0.03) | 0.80(0.08) | 0.40(0.06) | 0.38(0.03) | 0.91(0.01) | 0.90(0.03) |

---

> ### Author Response · Authors · 2024-11-25
> **Response to Reviewer 2GBK (3)**
>
> **Q4**: Including TACO with alpha=0 in Tables 2 and 3 might demonstrate its performance, as it would likely achieve the highest activity while maintaining high diversity, the primary differentiating factor in these studies.
>
>
> **A4**: In our updated offline MBO setting, the impact of the TFBS Reward can be thoroughly observed and analyzed. Specifically, we have included an ablation study on the effect of the TFBS Reward shown in Table 3. The "w/o TFBS Reward" setup corresponds to α=0, while our TACO model uses a default α=0.01. Additionally, we have also provided results for α=0.1 for further comparison.
>
> Incorporating the TFBS reward significantly enhances the Medium performance of TACO, achieving best results across all datasets. The method consistently outperforms the w/o TFBS Reward	 baseline by margins of 0.02, 0.01, and 0.02, respectively. These prior-informed rewards guide the policy to explore a more rational sequence space efficiently. Moreover, the biologically guided TFBS Reward is surrogate-agnostic, with the potential to achieve a similar effect to the regularization applied to surrogates in [4], by avoiding excessive optimization towards regions where the surrogate model gives unusually high predictions. The differences in the top fitness and diversity achieved by various models are relatively minor, with no consistent conclusion.
>
> Additionally, as the α increases from the default value of 0.01 to 0.1, our method shows improved performance in both Top and Medium metrics for K562 and SK-N-SH datasets. However, this improvement comes at the cost of a rapid drop in diversity. Interestingly, all metrics for the HepG2 dataset worsen as α grows. We hypothesize that this discrepancy arises from the TFBS Reward, precomputed using the LightGBM model, varying across datasets. Therefore, we recommend carefully tuning α in real-world scenarios to balance the trade-offs effectively.
>
> The updated ablation reuslts are in Section 4.4 Table 5 in the revised draft.
>
> **Table 3** Ablation study.
> | Dataset   | Setting              | Top ↑        | Medium ↑     | Diversity ↑     | Emb Similarity ↓ |
> |-----------|----------------------|--------------|--------------|-----------------|------------------|
> | **HepG2** | TACO (α = 0.01)      | **0.69 (0.03)** | **0.60 (0.05)** | **141.2 (1.92)** | 0.82 (0.05)       |
> |           | w/o Pretraining      | 0.68 (0.00)  | 0.55 (0.02)  | 139.4 (2.30)    | 0.69 (0.02)       |
> |           | w/o TFBS Reward      | 0.66 (0.05)  | 0.58 (0.07)  | 140.8 (1.64)    | **0.81 (0.05)**   |
> |           | α = 0.1              | 0.65 (0.06)  | 0.56 (0.08)  | 138.6 (3.21)    | 0.86 (0.04)       |
> | **K562**  | TACO (α = 0.01)      | 0.75 (0.09)  | 0.72 (0.10)  | 102.6 (20.14)   | 0.97 (0.04)       |
> |           | w/o Pretraining      | 0.66 (0.15)  | 0.59 (0.16)  | 103.6 (25.77)   | **0.83 (0.14)**   |
> |           | w/o TFBS Reward      | 0.76 (0.07)  | 0.71 (0.08)  | **106.2 (20.90)**| 0.94 (0.05)       |
> |           | α = 0.1              | **0.78 (0.01)** | **0.77 (0.01)** | 82.8 (4.02)     | **0.99 (0.00)**   |
> | **SK-N-SH** | TACO (α = 0.01)    | 0.68 (0.08)  |0.62 (0.08) | 121.4 (7.86)    | 0.90 (0.03)       |
> |           | w/o Pretraining      | 0.69 (0.02) | 0.57 (0.06)  | **131.8 (11.17)**| **0.74 (0.11)**   |
> |           | w/o TFBS Reward      | 0.67 (0.06)  | 0.60 (0.06)  | 111.6 (12.86)   | 0.89 (0.04)       |
> |           | α = 0.1              | **0.71** (0.01)  | **0.65** (0.02)  | 121.2 (5.45)    | 0.90 (0.05)       |
>
>
> **Q5**: Algorithm 1 appears too early in the paper.
>
> **A5**: Thank you for this suggestion. In the revised draft, we have moved Algorithm 1 to Appendix I.
>
> **Q6**: Equations 1, 3, and 4 could be moved to the appendix.
>
> **A6**: Thank you for the suggestion. We have moved Equation 4 to Appendix F in the revision. However, we have retained Equations 1 and 3 because, despite being common, they are crucial for clearly explaining our method and task—particularly since Equation 3 is directly related to Equation 2. Removing Equations 1 and 3 would also require other revisions to the main text. A Similar work also include these foundational equations for autoregressive models and reinforcement learning directly in the paper for better clarity and context [5].

---

> ### Author Response · Authors · 2024-11-25
> **Response to Reviewer 2GBK (4)**
>
> **Q7**: Could the authors provide more details on how the maximum number of steps T is determined?
>
> **Q8**: Could the authors clarify the following statement: "We set the maximum number of optimization iterations to 100, with up to 256 oracle calls allowed per iteration." Is one optimization iteration equivalent to one episode? If so, does "256 oracle calls" imply that T=256?
>
> **A7 & A8**: Thank you for pointing out these questions. We can address them together, as they are closely related. To address **Q8** first: One episode corresponds to the generation of a single fixed-length sequence (sequences of length 80 for yeast promoters or length 200 for human enhancers following the original dataset). In each optimization iteration, we generate 256 fixed-length sequences simultaneously, which corresponds to "256 oracle calls per iteration." After receiving feedback from the oracle, we update the policy once. This process is repeated for T = 100 iterations, resulting in a total of 256 multiplied by 100, which is 25,600 proposed sequences. The batch size is fixed at 256 in our setup, consistent with standard practices in baseline methods that process data in batches. Therefore, "256 oracle calls" reflects the batch size used in each iteration, not the total number of iterations. In summary, the number of iterations, represented by T, is 100, and the total oracle calls are determined by multiplying the batch size (256) with the number of iterations (100). To address **Q7**: Our setting is inspired by a molecular property optimization study [5], which focuses on SMILES sequences (a setup similar to ours as it also involves sequence-based optimization). In their study, due to the high likelihood of molecule duplication, the optimization process is terminated after achieving 25,000 unique oracle calls, even though the total number of oracle calls exceeds this threshold (40,000 in their case). However, Significant duplication is not observed in our experiments. As a result, we did not terminate early and instead set the total oracle calls to a comparable scale. By choosing 100 iterations, the total oracle calls become 256 multiplied by 100, resulting in 25,600. A detailed explanation has also been added to Section 4.1 in the revision.
>
> **Q9**: Why did the authors retrain HyenaDNA from scratch on D_star? Why not start from a pretrained HyenaDNA model?
>
> **A9**: Thanks for your question. We started with the pretrained weights of HyenaDNA and fine-tuned the model on relatively short CRE sequences to address the length discrepancy, as HyenaDNA was pretrained on much longer sequences. As shown in Table 4, fine-tuning on offline CRE data slightly improves performance. A detailed explanation has also been added to Section 3.2 and Appendix D.1 in the revision. We have also referenced Appendix D.2 in Section 4.1.
>
> **Table 4** Performance (hepg2 hard) comparison of pretrained and fine-tuned HyenaDNA on short CRE sequences.
>
> | Model                | Top ↑   | Medium ↑   |
> |----------------------|---------|------------|
> | Pretrained HyenaDNA  | 0.749   | 0.723      |
> | Fine-tuned HyenaDNA  | **0.751** | **0.729**  |
>
>
>
>
>
> **Q10**: The authors claim a lack of strong encoder backbones like ESM in the DNA field. This seems inaccurate, given recent publications like Nucleotide Transformer, DNABert, HyenaDNA, Caduceus, Evo, Borzoi, Enformer, and many others. If the authors disagree with the claims from these papers, they should provide a detailed argument.
>
> **A10**:  Our main point is to highlight that the embeddings output by the DNA language model are not universally generalizable. While there have been advancements in DNA language models, evidence suggests that they do not yet match the capabilities of models like ESM. Specifically:
> (1) ESM embeddings are known for their high versatility and are widely utilized in various downstream tasks, e.g., enzyme function prediction [6]. In contrast, as noted in [7], DNA foundation model embeddings often **perform no better than one-hot encodings**.
> (2) ESM’s language model head can achieve AUROC scores above 0.9 in pathogenic mutation prediction by directly calculating the log-likelihood ratio of reference and alternative alleles [8]. However, DNA foundation models currently perform significantly worse, with AUROC scores below 0.6 as reported in [9].
> (3) In addition to sequence-based DNA foundation models, some supervised DNA models have also been shown to exhibit limitations in distinguishing mutations across individuals [10] and recognizing long-range DNA interactions [11].
>
> We have included this discussion in Appendix D.2 in the revision. We have also referenced Appendix D.2 in Section 4.1.
>
> **Q11**: Using "medium" for both a dataset and a metric could be confusing.
>
> **A11**: Thank you for pointing this out. To avoid confusion between the dataset difficulty levels and the metric terminology, we have renamed "medium" (used for dataset difficulty) to "middle" in the revision.

---

> ### Author Response · Authors · 2024-11-25
> **Response to Reviewer 2GBK (5)**
>
> **Q12**: The authors frequently cite Almeida et al. but do not use their fly enhancer data in the evaluation. What motivated this decision?
>
> **A12**: We chose yeast promoters and human enhancers based on prior work by regLM [1], which provided well-documented datasets and preprocessing pipelines. This allowed us to focus on developing our method without spending excessive time on data preparation. As for Almeida et al.'s fly enhancer data [12], benchmarking remains limited aside from D3 [3], whose preprint does not provide reproducible code. Our attempt to train an oracle for fly enhancer expression using reglm’s pipeline yielded poor performance, with a Pearson correlation of 0.55 on the housekeeping subset—insufficient for further experiments. We plan to include fly enhancer data in future work. Training the fly enhancer oracle may require the "evolution-inspired data augmentations" mentioned in D3 [3], which we intend to explore in future work.
>
> **Q13**: At the end of the "conditional DNA generative models" section, the authors state: "However, these generative methods are designed to fit existing data distributions, limiting their ability to design sequences that have yet to be explored by humans." However, TACO seems to suffer from the same limitation, as its exploration space is bounded by the trained oracle, which is itself limited by the available data distribution. I would appreciate the authors' perspective on this.
>
> **A13**: Thank you for bringing up this important point. (1) First, we acknowledge that all models, including generative and optimization-based models, are inherently bounded by the oracle. What we aim to emphasize is that generative models, by design, are not optimized for exploring data that has yet to be observed or labeled (e.g., fitness for CRE). Generative models are trained to fit existing data distributions, which makes it challenging for them to efficiently interact with the oracle to incorporate newly labeled data. Directly fine-tuning generative models on new data often leads to mode collapse [14]. This limits their ability to explore beyond the observed data distribution effectively. (2) While, in theory, conditional generative models could condition on the maximum observed value to generate high-fitness sequences, in practice, their performance is constrained by the typically narrow and sparsely represented high-fitness regions in the data distribution. This limitation is well-demonstrated in our earlier response **A2** and Table 1, where we show that even regLM, which has access to a dataset containing higher-fitness values compared to the offline dataset, fails to generate sequences with fitness as high as those discovered by optimization-based methods. This highlights a key difference: optimization-based methods are better at targeting and expanding into high-fitness regions through iterative interaction with the oracle, whereas generative models often struggle to learn and exploit such distributions effectively.

---

> ### Author Response · Authors · 2024-11-30
> **Seeking Further Feedback Before Discussion Deadline**
>
> Dear Reviewer,
>
> Thank you for your constructive feedback on our paper. We greatly appreciate the time and effort you have dedicated to reviewing our work. In response to your comments, we have made the following key changes:
>
> - Added generative model baselines for comparison.
> - Introduced an embedding-based similarity metric to evaluate the diversity of the proposed sequences.
> - Provided more detailed ablations to demonstrate the contributions of our core components.
> - Addressed specific details and included a more comprehensive discussion about our work.
>
> As the discussion deadline is approaching, we kindly request further feedback from you to help us refine the quality of our paper and resolve any remaining concerns. Your insights are highly valued and will play a crucial role in enhancing the clarity and impact of this work.
>
> We understand that it is currently the Thanksgiving period (if applicable to you), and we apologize for any inconvenience caused by this message. We wish you a Happy Thanksgiving and sincerely thank you for your efforts in fostering progress within the research community.
>
> Best regards,
>
> The Authors

---

> > ### Comment · Reviewer_2GBK · 2024-11-30
> > **Response to authors' rebuttal**
> >
> > I warmly thank the authors for their rebuttal work.
> >
> > I apologize for the delay, I was indeed celebrating Thanksgiving.
> >
> > I have reviewed the updated paper, other authors' rebuttals, and all responses.
> >
> > - I appreciate the efforts made to address the comments from all reviewers. The updates have strengthened the paper.
> > - The model-based optimization discussion and experiment are particularly noteworthy.
> > - The addition of regLM and DDSM considerably strengthens the paper.
> > - The authors' efforts to introduce a new metric to measure diversity and the detailed explanations provided in response to my questions are appreciated.
> >
> > The authors have addressed most of my previous comments. However, I still have some reservations about the results:
> >
> > - A general concern is that the authors rank methods solely on average values without considering margins of error. TACO exhibits high variance. For instance, in the MBO experiment in Table 1, TACO shows the largest variance (except for the diversity metric with Adalead). The authors should not claim the superiority of one method over another if the differences lie within the margin of error, or they should provide statistical tests to support their claims.
> > - When discussing Table 1 (of the rebuttal, MBO results for human enhancers), the authors claim that "TACO outperforms the baselines on all datasets." However, if I compare TACO to regLM, for instance, regLM is within TACO's margin of error for the top metric and strongly outperforms it in the diversity and embedding similarity metrics. TACO outperforms regLM only for the medium metric. Given these results, it is unclear whether TACO offers a clear advantage over regLM.
> > - In Table 4 of the rebuttal (comparison of performance when pre-trained and fine-tuned), the authors report results with three significant digits without providing margins of error. This seems unwarranted given the observed variance in other experiments, and a maximum of two significant digits should be reported.
> > - In the ablation study (Table 3 of the rebuttal), the ablation that does not use the TBFS reward is, in most cases, within the margin of error of TACO with alpha=0.1. This supports my original comment that the TBFS reward does not appear to have a significant impact on performance.
> >
> > I thank the authors for their strong rebuttal work. I think that TACO is interesting and potentially valuable to the ICLR community. Therefore, I am increasing my score to 5.
> >
> > While TACO may not definitively outperform all baselines, or consistently benefit from the TBFS reward, the novelty of the method and the rigor of the research presented make it a valuable contribution. However, to increase my score further, I would expect the authors to either (1) provide statistical tests or further evidence to support their claims or (2) tone down their claims in the paper.

---

> > > ### Author Response · Authors · 2024-12-01
> > > **Follow-up to Reviewer Concerns (2)**
> > >
> > > **Table 5**. Hypothesis Test Results for TACO vs. regLM
> > >
> > > | Dataset  | Comparison       | Top (P-value) | Medium (P-value) |
> > > |----------|------------------|---------------|-------------------|
> > > | HepG2    | TACO > regLM     | **0.0000**    | **0.0000**        |
> > > | K562     | TACO > regLM     | **0.0000**    | **0.0000**        |
> > > | SK-N-SH  | TACO > regLM     | **0.0000**    | **0.0000**        |
> > >
> > > **Q3**: In Table 4 of the rebuttal (comparison of performance when pre-trained and fine-tuned), the authors report results with three significant digits without providing margins of error. This seems unwarranted given the observed variance in other experiments, and a maximum of two significant digits should be reported.
> > >
> > > **A3**: Thank you for pointing this out. We acknowledge the imprecision. The results presented in round 1 were derived from an earlier setting in Section 4.2 using a single random seed (HepG2 hard). We have now updated the evaluation with a more comprehensive analysis, utilizing 5 random seeds to compare the official pretrained HyenaDNA model and the fine-tuned model on CRE data (HepG2 offline MBO). The results, as shown in Table 6, indicate that fine-tuning on CRE sequences achieves slightly better performance, although the improvement appears modest. This enhancement is likely attributable to addressing the pretraining-to-CRE sequence length gap. We plan to incorporate the results from this table into the revised draft as an update to Appendix D.1 Table 8.
> > >
> > > **Table 6**. Performance (HepG2 offline MBO) comparison of pretrained and fine-tuned HyenaDNA on short CRE sequences.
> > >
> > > | Model                | Top ↑         | Medium ↑      | Diversity ↑  |
> > > |----------------------|---------------|---------------|--------------|
> > > | Pretrained HyenaDNA  | 0.68 (0.02)   | 0.58 (0.03)   |  140.8 (0.84) |
> > > | Fine-tuned HyenaDNA  | **0.69** (0.03) | **0.60** (0.05) | **141.2** (1.92)  |
> > >
> > >
> > >
> > > **Q4**: In the ablation study (Table 3 of the rebuttal), the ablation that does not use the TBFS reward is, in most cases, within the margin of error of TACO with alpha=0.1. This supports my original comment that the TBFS reward does not appear to have a significant impact on performance.
> > >
> > > **A4**: Thank you for your detailed review and for highlighting this concern. To address it, we conducted hypothesis tests to evaluate whether `alpha=0.01` or `alpha=0.1` outperforms `alpha=0`. As with Table 4 in the rebuttal, we report these results based on hypothesis tests conducted on sample-level data.
> > >
> > > Shown in Table 7, the results demonstrate that the TFBS reward (`alpha=0.01` or `alpha=0.1`) significantly improves performance in 5 out of 6 Medium comparisons, showcasing its effectiveness for this metric. However, for the Top metric, significant improvements are observed in only 2 out of 6 comparisons. This suggests that while the TFBS reward does have some impact on top-performing sequences, its primary effect lies in improving the broader fitness distribution represented by the Medium metric. The conclusion here aligns with our current draft (Lines 502-505). To make the results of our ablation study more convincing, we plan to include Table 5 in the appendix of the final draft and reference it in Section 4.4.
> > >
> > > **Table 7**. Hypothesis Test Results for the Effect of TFBS Reward
> > > | Dataset  | Comparison                  | Top (P-value) | Medium (P-value) |
> > > |----------|-----------------------------|---------------|-------------------|
> > > | HepG2    | (alpha=0.01) > (alpha=0)   | **0.0104**    | **0.0000**        |
> > > |          | (alpha=0.1) > (alpha=0)    | 0.9624        | **0.0063**        |
> > > | K562     | (alpha=0.01) > (alpha=0)   | 0.6769        | 0.1110            |
> > > |          | (alpha=0.1) > (alpha=0)    | 0.8842        | **0.0000**        |
> > > | SK-N-SH  | (alpha=0.01) > (alpha=0)   | 0.1658        | **0.0003**        |
> > > |          | (alpha=0.1) > (alpha=0)    | **0.0000**    | **0.0000**        |

---

> > > > ### Comment · Reviewer_2GBK · 2024-12-02
> > > >
> > > > Thank you for providing this additional information. This effectively addresses the remaining concerns I had, and I'm happy to raise my score from 5 to 8.

---

> > > > > ### Author Response · Authors · 2024-12-02
> > > > > **Grateful Acknowledgment**
> > > > >
> > > > > We sincerely thank you for your valuable feedback and thoughtful suggestions, which have significantly enhanced the quality of our paper. Your recognition of our work and your generous decision to raise your score mean a great deal to us.

---

> ### Author Response · Authors · 2024-12-01
> **Follow-up to Reviewer Concerns (1)**
>
> First of all, we sincerely apologize for disturbing your Thanksgiving and hope you had a wonderful Thanksgiving again.
>
> Thank you very much for your further feedback, which has been extremely helpful in improving our paper. We have provided additional explanations and conducted further experiments to make our claims more robust and precise.
>
>
> **Q1**:  A general concern is that the authors rank methods solely on average values without considering margins of error. TACO exhibits high variance. The authors should not claim the superiority of one method over another if the differences lie within the margin of error, or they should provide statistical tests to support their claims.
>
> **Q2**: However, if I compare TACO to regLM, for instance, regLM is within TACO's margin of error for the top metric and strongly outperforms it in the diversity and embedding similarity metrics. TACO outperforms regLM only for the medium metric. Given these results, it is unclear whether TACO offers a clear advantage over regLM.
>
> **A1 & A2**: Thank you for your detailed review of the experimental data and for highlighting potential issues.
>
> To address your concerns:
>
> - We acknowledge that our initial evaluation relied primarily on the mean of the metrics to compare models, which overlooked the large standard deviation exhibited by our method. This standard deviation likely arises from the fine-tuning paradigm of AR models, where each episode begins with sequences generated from scratch, making the optimization process highly dependent on the fitness of the initial proposed sequences. Notably, this large standard deviation is primarily observed in Section 4.3 (offline MBO setting) and not in Section 4.2. As shown in the ablation study, the standard deviation remains significant even without pretraining or TFBS rewards.
>   **Planned Addition to Section 4.3 (Line 456):**
>   *"However, TACO exhibits a higher standard deviation. This standard deviation likely stems from the fine-tuning paradigm of AR models, where each episode begins with sequences generated from scratch, causing the optimization process to be highly dependent on the fitness of the initially proposed sequences."*
>
> - The claim that *"TACO outperforms the baselines on all datasets"* may have been an oversight on our part. We primarily intended to make this claim in the context of comparing fitness-related metrics between TACO, regLM, and DDSM. In **A2** of the round 1 rebuttal, we acknowledge that the rebuttal could have led to ambiguity, and we have revised **A2** of the round 1 rebuttal to emphasize that the comparison was focused on generative models with the fitness metric, while also recognizing that generative models such as regLM and DDSM perform better on diversity metrics. Furthermore, in the main paper, we never stated that *"TACO outperforms the baselines on all datasets"*. To clarify, we have decided to explicitly highlight in the draft (Line 459) that generative models like regLM and DDSM achieve better diversity.
>
>
> - We also recognize the reviewer's concern regarding the lack of clear superiority of our method over regLM. To address this, we conducted hypothesis testing to evaluate whether TACO significantly outperforms regLM on fitness-related metrics. Performing statistical tests directly on the aggregated metrics (Top and Medium) was deemed inappropriate due to the limited variability caused by the small number of random seeds (5 independent runs per condition). Instead, we conducted the tests on the generated data samples, which provided a larger pool of data points for more robust evaluation. For this analysis, we used sequences generated in previous runs. For each seed, the top 16 highest fitness values were combined to form the Top subset, and the top 128 values were combined to form the Medium subset. Across all 5 seeds, this resulted in 80 data points for the Top subset and 640 for the Medium subset per condition. Statistical tests were performed on these subsets to ensure sufficient sample size and statistical power. We applied the one-sided Mann-Whitney U test to determine whether the results for TACO were significantly greater than those for regLM. **Bold** values in the table below indicate statistically significant results (\(P < 0.05\)). Shown in Table 5, these results confirm that while RL fine-tuning may reduce diversity compared to regLM, it significantly enhances both Top and Median fitness.

---

### Author Response · Authors · 2024-11-25
**General Response**

Dear Reviewers,

We sincerely appreciate your valuable time and insightful feedback. Since your comments were detailed and highly valuable, we have revised the draft accordingly and uploaded the updated version, with all changes highlighted in **blue** for clarity. Below is a summary of the main revisions:

1. **Offline model-based optimization (MBO) discussion and experiments (Section 4.3, Appendix M, Table 4, Table 12-16)**
In response to Reviewer BjdH's suggestion, we have added offline MBO [1] experiments. Briefly, we used the oracle previously trained on the full dataset, but to avoid directly optimizing the oracle, we trained a surrogate model on a subset to guide the optimization process. **All the additional experiments conducted during the rebuttal phase** are based on this offline MBO setting unless otherwise specified., where optimization is performed using the surrogate model, and evaluation is done with the oracle. Apart from using the surrogate to guide the optimization process, all other settings remain consistent with the previous ones.

2. **Additional baselines and metric (Section 4.3, Table 4, Appendix M)**
We have added the conditional generative models regLM [2] and DDSM [3] as baselines. Additionally, inspired by D3 [4], we introduced a new metric, **Emb Similarity**, which measures sequence diversity based on the pairwise similarity of embeddings generated by the oracle.

3. **Relocation of Content to Appendix (Appendix I, Appendix K, Table 11)**
We moved the algorithm flowchart to Appendix I and the discussion of motif-based machine learning from the related work section to Appendix K. Additionally, most of the results related to yeast promoters were relocated to Appendix M, as these experiments provide relatively low information content. The ablation experiments supporting the effectiveness of supporting RL designs have been moved to Appendix I.2.

4. **Factual Error Fixes (Figure2, Figure 3, Figure 5, Section 4.1)**
We have corrected inaccuracies in the experimental details and descriptions of figures.

5. **Enhanced Ablation Studies on Core Contributions (Section 4.4, Table 5)**
We conducted more detailed ablation studies on the two core contributions: Pretraining and TFBS Reward, providing deeper insights into the role of each component.

6. **More Extensive Discussion (Section 6, Appendix D.2, Appendix J, Appendix L)**
Additional discussions have been added regarding existing works and potential improvements to the method.

**References**

[1] Reddy, Aniketh Janardhan, et al. "Designing Cell-Type-Specific Promoter Sequences Using Conservative Model-Based Optimization." *bioRxiv* (2024).

[2] Lal, Avantika, et al. "regLM: Designing realistic regulatory DNA with autoregressive language models." *International Conference on Research in Computational Molecular Biology.* Cham: Springer Nature Switzerland, 2024.

[3] Avdeyev, Pavel, et al. "Dirichlet diffusion score model for biological sequence generation." *International Conference on Machine Learning.*

[4] Sarkar, Anirban, et al. "Designing DNA With Tunable Regulatory Activity Using Discrete Diffusion." *bioRxiv*.

---

> ### Author Response · Authors · 2024-11-28
> **General Response (2)**
>
> Dear Reviewers,
>
> We sincerely appreciate your valuable time and insightful feedback. As the revision submission deadline approaches, we have made further updates to our draft. The new version has been uploaded, and all the changes are highlighted in **orange** for your convenience. Below is a summary of the main revisions:
>
> 1. **Added discussions on the contributions of related work (Section 2)**
> We have primarily enhanced the discussion in Section 2 to elaborate on the contributions of [1] and [2], while emphasizing how our work builds upon and differs from theirs.
>
> 2. **Added discussions on the Gradient Ascent method (Section 4.1, Appendix N)**
> We discussed in Section 4.1 that the main reason we do not compare with Gradient Ascent methods is that our approach does not rely on a differentiable surrogate. Additionally, in Appendix N, we analyzed the performance of Gradient Ascent and fairly presented the results of TACO and baselines under the 60th percentile offline MBO setting.
>
> We greatly appreciate the reviewer's valuable feedback. If you have any further concerns, please raise them in the forum, and we will actively address them. Your input will undoubtedly help enhance the quality of our paper.
>
>
> **References**
>
> [1] Reddy, Aniketh Janardhan, et al. "Designing Cell-Type-Specific Promoter Sequences Using Conservative Model-Based Optimization." *bioRxiv*. 2024.
>
> [2] Angermueller, Christof, et al. "Model-based reinforcement learning for biological sequence design." *International conference on learning representations.* 2019.

---

### Meta-Review · Area_Chair_jate · 2024-12-23

**Metareview:**

This paper presents TACO, a reinforcement learning-based framework for designing cis-regulatory elements (CREs), incorporating pretrained autoregressive models and transcription factor binding site (TFBS) rewards to guide sequence optimization. Strengths indicated by the reviewers include the clear writing and the novelty of integrating RL fine-tuning with TFBS-driven rewards. There were many different criticisms, which were addressed in the rebuttal. I thus believe this paper is now ready to be accepted.

**Additional Comments On Reviewer Discussion:**

Criticisms included over-reliance on oracle-based evaluation and insufficient comparisons with state-of-the-art methods. The authors addressed these concerns by incorporating offline model-based optimization (MBO) experiments, adding baseline comparisons with models like regLM and DDSM. During the rebuttal period, Reviewer 2GBK first increased to 5 and then upon further rebuttal increased to 8. Reviewer BjdH was unresponsive and I thus did not take their vote into account as much.

---

### Decision · Program_Chairs · 2025-01-22

Accept (Poster)